

**Effects of model resolution and parameterizations on the simulations of clouds, precipitation,**
**and their interactions with aerosols**
Seoung Soo Lee[1], Zhanqing Li[1], Yuwei Zhang[1], Hyelim Yoo[2]
[1]Earth System Science Interdisciplinary Center, University of Maryland, College Park, Maryland
[2]Earth Resources Technology, Inc., National Oceanic and Atmospheric Administration, College Park,
Maryland
Corresponding author: Seoung Soo Lee
Office: (303) 497-6615
Cell: (609) 375-6685
Fax: (303) 497-5318
E-mail: cumulss@gmail.com,slee1247@umd.edu



**Abstract**
This study investigates the effects of model resolution and microphysics parameterizations on the
uncertainties or errors in the simulations of clouds, precipitation, and their interactions with aerosols
using the Global Forecast System (GFS) model as one of the representative numerical weather
prediction (NWP) models. For this investigation, we used the GFS model results and compare them
with those from the cloud-system resolving model (CSRM) simulations as benchmark simulations that
adopt high resolutions and full-fledged microphysical processes. These simulations were evaluated
against observations and this evaluation demonstrated that the CSRM simulations can function as
benchmark simulations. Substantially lower updrafts and associated cloud variables (e.g., cloud mass
and condensation) were simulated by the GFS model compared to those simulated by the CSRM. This
is mainly due to the coarse resolutions in the GFS model. This indicates that the parameterizations that
represent sub-grid processes in the GFS model do not work well and thus need to be improved. Results
here also indicate that the use of the coarse resolution in the GFS model lowers the sensitivity of
updrafts and cloud variables to increasing aerosol concentrations compared to the CSRM simulations.
The parameterization of the saturation process plays an important role in the sensitivity of cloud
variables to aerosol concentrations while the parameterization of the sedimentation process has a
substantial impact on how cloud variables are distributed vertically. The variation in cloud variables
with resolution is much greater and contributes to the discrepancy between the GFS and CSRM
simulations to a much greater degree than what happens with varying microphysics parameterizations.



## 1. Introduction

The treatment of clouds and precipitation and their interactions with aerosols in the NWP models is likely a major source of errors in the simulations of the water and energy cycles (Sundqvist et al., 1989; Randall et al., 2006; Seifert et al., 2012). The NWP community has recognized that the accurate representation of clouds, precipitation, and cloud-aerosol-precipitation interactions (CAPI) is important for the improvement of the NWP models and thus, some of these models have started to improve the representation by considering CAPI (Morcrette et al., 2011; Sudhakar et al., 2016).

CAPI may not have a substantial impact on the total precipitation amount but they do affect the temporal and spatial variabilities of precipitation (Li et al., 2011; van den Heever et al., 2011; Seifert et al., 2012; Lee and Feingold, 2013; Fan et al., 2013; Lee et al., 2014), whose importance increases as the temporal/spatial scales of forecast decrease. The distribution of extreme precipitation events such as droughts and floods, closely linked to the spatiotemporal variability, has important social and economic implications.

In recent years, resolutions in the NWP models have increased to the point that the traditional cumulus parameterization schemes may no longer work properly. Motivated by this, scale-aware cumulus parameterization schemes (e.g., Bogenschutz and Krueger, 2013; Thayer-Calder et al., 2015; Griffin and Larson, 2016) are being implemented into these models of different resolutions for better representation of clouds and precipitation.



The uncertainties or errors in the simulations of clouds, precipitation, and CAPI in the NWP
models may be incurred both from microphysics parameterizations and from model resolutions. The
implementation of the cloud microphysics such as the two-moment (e.g. Morrison and Gettelman, 2008;
Morrison et al., 2009) and scale-aware schemes are intended to reduce these uncertainties. It is
important to first understand and quantify the uncertainties associated with the two-moment scheme and
how model resolutions create the uncertainties, as well as the relative significance between the
uncertainties associated with the two-moment scheme and those created by the resolutions. These
understanding and quantification can provide us with a guideline on how to represent microphysics in
the two-moment schemes and sub-grid processes in the scale-aware schemes for the efficient reduction
of the uncertainties in the NWP models. Note that the representation of sub-grid processes requires
information on the contribution of resolution to the uncertainties and, in this study, we focus on the two-
moment scheme developed by Morrison and Gettelman (2008) and Morrison et al. (2009), which is
referred to as the MG scheme, henceforth.
Fan et al. (2012) and Khain et al. (2015) have shown that the parameterizations of three key
microphysical processes (i.e., saturation, collection, and sedimentation) in microphysical schemes act as
a main source of errors in the simulation of clouds, precipitation, and CAPI. We try to identify and
quantify the errors or the uncertainties through comparisons between simulations with
parameterizations of the three key processes in the MG scheme and the CSRM simulations with full-
fledged microphysical processes. Regarding the understanding of the uncertainties arising from the
choice of resolution, we also perform comparisons between the high-resolution CSRM simulations and



the low-resolution simulations, and do additional comparisons with the GFS simulations. This helps
gain an understanding of how the microphysical representation and coarse resolution in the GFS model
(as compared to those in the CSRM) contribute to the uncertainties in the GFS simulations of clouds
and precipitation by accounting for CAPI. Here, the CSRM simulations act as benchmark simulations
by representing microphysical processes with high-level sophistication and by resolving cloud-scale
physical and dynamic processes with high resolutions.

**2. Models**

**2.1 The CSRM**

The Advanced Research Weather Research and Forecasting (ARW) model, a non-hydrostatic
compressible model, is the CSRM selected for use in this study. A fifth-order monotonic advection
scheme is used for the advection of cloud variables (Wang et al., 2009). The ARW model considers
radiation processes by adopting the Rapid Radiation Transfer Model for General Circulation Models
(RRTMG) (Fouquart and Bonnel, 1980; Mlawer et al., 1997). The RRTMG considers the effects of
aerosols on the effective sizes of hydrometeors and the associated changes in radiation.
For an assessment of the uncertainties in the MG scheme, which is a type of a bulk scheme, we
need to use microphysics schemes that are much more sophisticated than the MG scheme. Through
extensive comparisons between various types of bin schemes and bulk schemes, Fan et al. (2012) and



Khain et al. (2015) have concluded that the use of bin schemes or bin-bulk schemes is desirable for
reasonable simulations of clouds, precipitation, and their interactions with aerosols. This is because
these schemes do not use a saturation adjustment, a mass-weight mean terminal velocity, or constant
collection efficiencies that have been used in bulk schemes. Instead, bin schemes use predicted
supersaturation levels, and terminal velocities and collection efficiencies that vary with the sizes of
hydrometeors. Based on the work by Fan et al. (2012) and Khain et al. (2015), this study considers bin
schemes to be a full-fledged microphysics schemes against which the uncertainties in the MG scheme
can be assessed. Hence, a bin scheme is adopted in the CSRM used here.
The bin scheme adopted by the CSRM is based on the Hebrew University Cloud Model described
by Khain and Lynn (2009). The bin scheme solves a system of kinetic equations for the size distribution
functions of water drops, ice crystals (plate, columnar and branch types), snow aggregates, graupel and
hail, as well as cloud condensation nuclei. Each size distribution is represented by 33 mass-doubling
bins, i.e., the mass of a particle $m_k$ in the $k^{th}$ bin is $m_k = 2m_{k-1}$.
It is assumed that aerosol particles are composed of ammonium sulfate. The aerosol size
distribution is calculated prognostically with sinks and sources, which include advection, droplet
nucleation, and aerosol regeneration from droplet evaporation (Fan et al., 2009). Aerosol activation is
calculated according to the Köhler theory, i.e., aerosol particles with radii exceeding the critical value at
a grid point are activated to become droplets based on predicted supersaturation, and the corresponding
bins of the aerosol spectra are emptied. After activation, the aerosol mass is transported within
hydrometeors by collision-coalescence and removed from the atmosphere once hydrometeors that



contain aerosols reach the surface. Aerosol particles return to the atmosphere upon evaporation or the
sublimation of hydrometeors that contain them.

**2.2 The GFS model**

The GFS model is a global NWP model that is run by the National Oceanic and Atmospheric
Administration (NOAA). The GFS model has 64 vertical sigma-pressure hybrid layers and a T382 (~ 35
km) horizontal resolution. Output fields for a forecast generated at 3-hour intervals (i.e. at 03, 06, 09, 12,
15, 18, 21, 24 universal coordinated time, or Z), starting from the control time of 00Z, are used for this
study.
The GFS model posts parameters for 21 vertically different layers. From the surface (1000 hPa) to
the 900-hPa level, the vertical resolution is 25 hPa. Less than 900 hPa, there are 16 levels at a 50-hPa
resolution up to 100 hPa. The cloud phase is determined by the mean temperature ($T_c$) of a cloud layer
which is defined as the average of temperatures at the top and bottom of a cloud layer. If $T_c$ is less than
258.16 K, the cloud layer is an ice cloud; otherwise, it is a water cloud.
A prognostic condensate scheme by Moorthi et al. (2001) has been used to parameterize clouds in
the GFS model. In this scheme, cloud mass, one of the representative cloud variables, evolves by
considering the cloud-mass advection, diffusion and conversion to precipitation, and the diagnosed sub-
grid and grid-scale condensation and evaporation. The grid-scale condensation and evaporation are
calculated based on Sundqvist et al. (1989) and Zhao and Carr (1997), while the sub-grid condensation



and evaporation are calculated based on a cumulus parameterization that adopts the mass-flux
approach. This cumulus parameterization was developed by Moorthi et al. (2001) based on a simplified
Arakawa-Schubert scheme (Pan and Wu, 1995).

**3. The cases**

**3.1 The Seoul case**

A mesoscale convective system (MCS) was observed over Seoul, Korea (37.57$^o$N, 126.97$^o$E; 0900 local
solar time (LST) 26 July 2011–0900 LST 27 July 2011). This case, referred to as the Seoul case,
involved heavy rainfall with a maximum precipitation rate of ~150 mm h$^{-1}$. This heavy rainfall caused
flash floods and landslides on a mountain at the southern flank of the city, leading to the deaths of 60
people.
At 0900 LST July 26th 2011, favorable synoptic-scale features for the development of heavy
rainfall over Seoul were observed. The western Pacific subtropical high (WPSH) was located over the
southeast of Korea and Japan, and there was a low-pressure trough over north China (Figure 1a). Low-
level jets between the flank of the WPSH and the low-pressure system brought warm, moist air from the
Yellow Sea to the Korean Peninsula (Figure 1b). Transport of warm and moist air by the southwesterly
low-level jet is an important condition for the development of heavy rainfall events over Seoul (Hwang
and Lee, 1993; Sun and Lee, 2002).





### 3.2 The Houston case



An MCS was observed over Houston, Texas (29.42$^{o}$N, 94.45$^{o}$W; 0700 LST 18 July 2013–0400 LST 19
July 2013). The Houston case involved moderate rainfall with a maximum precipitation rate of ~50 mm
h$^{-1}$.
At 0500 LST, two hours before the initiation of convection, the low-level wind in and around
Houston was southerly (Figure 1c), favoring the transport of water vapor from the Gulf of Mexico to
the Houston area. Associated with this, the environmental convective available potential energy (CAPE)
(Figure 1d) in and around Houston along the coastline was high (as represented by red areas in Figure
1d). The high CAPE provided a favorable condition for the development of the MCS.

**4. Simulations**

**4.1 The CSRM simulations**

Using the ARW model and its bin scheme, a three-dimensional CSRM simulation of the observed MCS
was performed over the MCS period for each of the cases.
Initial and boundary conditions for the control run are derived from the National Centers for
Environmental Prediction GFS final (FNL) analysis. All experiments employ a prognostic surface skin



temperature scheme (Zeng and Beljaars, 2005) and a revised roughness length formulation (Donelan
et al., 2004).
The control run for each of the cases consists of a domain with a Lambert conformal map
projection. The domain is marked by the rectangle for the Seoul case in Figure 2a and the domain for
the Houston case is shown in Figure 2b. While the control run for the Seoul case is referred to as "the
control-Seoul run", the control run for the Houston case is referred to as "the control-Houston run",
henceforth. The domain for the Seoul (Houston) case covers the Seoul (Houston) area and to resolve
cloud-scale processes, a 500-m horizontal resolution is applied to the domain. The domain has 41
vertical layers with resolutions ranging from 70 m near the surface to 800 m at the model top (~50 hPa).
Note that the cumulus parameterization scheme is not used in this domain where convective rainfall
generation is assumed to be explicitly resolved. Based on observations, the aerosol concentration at the
surface at the first time step is set at 5500 (1500) $cm^{-3}$ for the Seoul (Houston) case. Above the top of
the planetary boundary layer (PBL) around 2 km, the aerosol concentration reduces exponentially.
To examine and isolate CAPI, i.e., the effect of increasing the loading of aerosols on clouds and
precipitation, the control run is repeated with the aerosol concentration at the first time step reduced by
a factor of 10. This factor is based on observations showing that that reduction in aerosol loading
between polluted days and clean days is generally tenfold over Seoul and Houston(Lance et al., 2009;
Kim et al., 2014). This simulation is referred to as the low-aerosol-Seoul run for the Seoul case and the
low-aerosol-Houston run for the Houston case. Since the control-Seoul run and the control-Houston run
involve higher aerosol concentrations than the low-aerosol-Seoul run and the low-aerosol-Houston run,



respectively, for naming purposes, the control-Seoul run and the control-Houston run are also
referred to as the high-aerosol-Seoul run and the high-aerosol-Houston run, respectively.
In addition to the simulations described above, more simulations were performed to fulfill the goals
of this study (Table 1). Details of those simulations are given in the following sections.

**4.2 The GFS simulations**

Note that the GFS produces the forecast data over the globe and for this study, we use the data
only during the MCS period and only at grid points in the domain for each case. We collect GFS data in
the domain and then average the data over those grid points at each of the GFS time steps for each of
the cases. For the comparison between the GFS and CSRM simulations at specific time steps over the
MCS period, these averaged data are compared to the CSRM simulations during the period. In case the
time and domain-averaged GFS data are compared to the CSRM counterparts, these averaged data are
averaged again over the MCS period and compared to their CSRM counterparts.

**5. Results**

**5.1 Test on the effects of resolution on the simulations of clouds,**

**precipitation, and CAPI**




### 5.1.1  Cloud liquid content (CLC) and cloud ice content (CIC)

To test the effects of resolution on the simulations of clouds, precipitation, and their interactions with aerosols, we repeat the standard CSRM runs at the 500-m resolution (i.e., the high-aerosol-Seoul run, the low-aerosol-Seoul run, the high-aerosol-Houston run, and the low-aerosol-Houston run) by using 15- and 35-km resolutions instead. These resolutions are similar to those generally adopted by current NWP models (e.g., the GFS model) and thus comparisons between these repeated simulations and the CSRM simulations can evaluate how coarse resolutions adopted by the NWP models affect the simulations of clouds, precipitation, and their interactions with aerosols. The repeated simulations at the 15-km resolution are referred to as the high-aerosol-15-Seoul run, the low-aerosol-15-Seoul run, the high-aerosol-15-Houston run, and the low-aerosol-15-Houston run, while the repeated simulations at the 35-km resolution are referred to as the high-aerosol-35-Seoul run, the low-aerosol-35-Seoul run, the high-aerosol-35-Houston run, and the low-aerosol-35-Houston run. In this study, simulations whose name includes "high-aerosol" represent the polluted scenario, while those whose name includes "low-aerosol" represent the clean scenario. In the following, we describe results from the standard and repeated simulations. For the Houston case, no clouds form at the 35-km resolution, so the description of results is only done for results at the 15-km resolution.

Figures 3a and 3b show the vertical distributions of the time- and domain-averaged CLC in the simulations for the Seoul case and the Houston case, respectively. Figures 4a and 4b show the vertical distributions of the time- and domain-averaged CIC in the simulations for the Seoul case and the



Houston case, respectively. There are increases in the cloud mass (represented by CLC and CIC) with
increasing aerosol concentration between the polluted scenario and the clean scenario not only for both
the Seoul and Houston cases but also at all resolutions considered. There are substantial decreases in the
cloud mass at the 15- and 35-km resolutions compared to the cloud mass in the simulations at the 500-
m resolution. In addition, increases in the cloud mass with increasing aerosol concentration reduce
substantially as the resolution coarsens. At the 500-m resolution, on average, there is about a ~30–50%
increase in cloud mass, while at the 15- or 35-km resolutions, there is only a ~2–5% increase in cloud
mass in both cases.
For both the Seoul and Houston cases, comparisons between the cloud mass produced by the
GFS simulations and that produced by the ARW simulations show that the GFS-simulated cloud mass
is similar to that in the ARW simulations at the 15- and 35-km resolutions. However, the GFS-
simulated cloud mass is much smaller than that in the ARW simulations at the 500-m resolution, i.e.,
the CSRM simulations. This suggests that the coarse resolutions used in the GFS simulations are an
important cause of the differences in cloud mass between the CSRM simulations and the GFS
simulations.

**5.1.2 Liquid water path (LWP) and ice water path (IWP)**

Figures 5a and 5b show the time series of the domain-averaged LWP and IWP for the Seoul case while
Figures 6a and 6b show the same for the Houston case. Note that LWP and IWP are the vertical



integrals of CLC and CIC, respectively. Consequently, the same behavior as that of CLC and CIC is
seen, namely, there are increases in LWP and IWP with increasing aerosol concentrations between the
polluted and clean scenarios at all resolutions, while there are decreases in LWP and IWP with the use
of the 15- and 35-km resolutions compared to using the 500-m resolution. Also, the sensitivity of LWP
and IWP to increasing aerosol concentrations reduces significantly as the resolution coarsens.
In Figures 5 and 6, satellite-observed LWP and IWP for both cases follow reasonably well their
CSRM-simulated counterparts for the polluted scenario. This shows that the CSRM simulations
perform well and can thus represent benchmark simulations. The GFS-produced LWP and IWP are
similar to those in the ARW simulations at the 15- and 35-km resolutions and are much smaller in
magnitude than those from the CSRM simulations and observations. Hence, the discrepancy in LWP
and IWP between the GFS and CSRM simulations or that between the GFS simulations and
observations is closely linked to the coarse resolution adopted by the GFS simulations. Taking the
CSRM simulations as benchmark simulations, we see that GFS simulations underestimate the cloud
mass compared to observations mainly due to the coarse resolution adopted by the GFS model.
Among the ARW simulations, the sensitivity of the cloud mass to increasing aerosol
concentrations reduces considerably with coarsening resolution. CSRM simulations are benchmark
simulations so the sensitivity in the CSRM simulations is the benchmark sensitivity. Note that the GFS
simulation results and the ARW simulations at coarse resolutions of 15 and 35 km are similar. Their
sensitivities are thus also likely similar, i.e., the sensitivity of the cloud mass to increasing aerosol



concentrations in the GFS simulation is likely to be underestimated compared to the benchmark
sensitivity of the CSRM simulations.

### 5.1.3   Updrafts, condensation, and deposition

To understand the response of the cloud mass to increasing aerosol concentrations, and the variation in
the cloud mass and its response to increasing aerosol concentrations with varying resolutions as shown
in Figures 3, 4, 5, and 6, we calculate updraft mass fluxes since these fluxes control supersaturation that
in turn controls condensation and deposition as key determination factors for the cloud mass. We also
obtain condensation and deposition rates. The vertical distributions of time- and domain-averaged
updraft mass fluxes, condensation rates, and deposition rates for the Seoul and Houston cases are shown
in Figures 7, 8, and 9, respectively.
As seen for the cloud mass, updraft mass fluxes, and condensation and deposition rates increase
with increasing aerosol concentrations between the polluted scenario and the clean scenario at all
resolutions and for all cases considered. Aerosol-induced percentage increases in updraft mass fluxes,
and deposition and condensation rates at the 500-m resolution are approximately one order of
magnitude greater than those at the 15-km and 35-km resolutions. Stated differently, the sensitivity of
updraft mass fluxes to increasing aerosol concentrations reduces substantially with coarsening
resolution and due to this, the sensitivity of deposition and condensation rates, and thus the cloud mass,
to   increasing   aerosol   concentrations   also   reduces   substantially   with   coarsening   resolution.



Similar to the situation with the cloud mass, the GFS-produced updraft mass fluxes are much smaller
than those produced by the ARW simulations at the 500-m resolution (or the CSRM simulations) and
similar to those produced by the ARW simulations at the 15- and 35-km resolutions (Figure 7). Hence,
the discrepancy in updraft mass fluxes between the GFS simulations and the CSRM simulations is
closely linked to the discrepancy in resolutions between these two types of simulations. The
underestimation of the updraft mass fluxes in the GFS simulations is mainly due to the coarse resolution
adopted by the GFS model. Taking the sensitivity of updraft mass fluxes to increasing aerosol
concentrations in the CSRM simulations as the benchmark sensitivity, the GFS simulations likely also
underestimate the sensitivity.
Figure 10 shows the frequency distribution of updrafts over the updraft speed, which is normalized
over the domain and the simulation period. We first calculate the frequency over the domain at each
time step and in each discretized updraft bin. The frequency in each bin and at each time step is then
divided by the total number of grid points in the whole domain. The normalized frequency at each time
step is summed over all of the time steps in each updraft bin. This sum is divided by the total number of
time steps as the final step in the normalization process. With coarsening resolution, the normalized
frequency of weak updrafts with speeds less than ~2 m s$^{-1}$ increases for both scenarios in both cases.
However, the normalized frequency of strong updrafts with speeds greater than ~2 m s$^{-1}$ reduces with
coarsening resolution. The frequency shift from high-level updraft speeds to low-level speeds leads to a
reduction in mean updrafts with coarsening resolution for both scenarios in both cases.



The updraft frequency is greater in the polluted scenario than in the clean scenario at all resolutions and for all cases. The overall difference in the frequency between the scenarios reduces with coarsening resolution. This is associated with the reduction in the sensitivity of the averaged updrafts to increasing aerosol concentrations with coarsening resolution. In particular, the difference in the frequency for weak updrafts (speeds less than ~2 m s$^{-1}$) between the scenarios does not vary much with coarsening resolution. On average, the percentage difference for weak updrafts is less than 2–3% at all resolutions. However, the difference for strong updrafts varies significantly with varying resolution. The mean difference for strong updrafts varies from ~30–60% for the 500-m resolution to less than ~5–6 % for the 15- and 35-km resolutions. Numerous studies (e.g., Khain et al., 2005; Seifert and Beheng, 2006; Tao et al., 2007, 2012; van den Heever and Cotton, 2007; Storer et al., 2010; Lee et al., 2013, 2017) have shown that aerosol-induced invigoration of convection through aerosol-induced increases in freezing or aerosol-induced intensification of gust fronts is the main mechanism behind aerosol-induced increases in updraft mass fluxes or the intensity of updrafts. Based on this, analyses of the updraft frequency here suggest that strong updrafts are more sensitive to aerosol-induced invigoration than weak updrafts. The variation in the sensitivity of the averaged updrafts to increasing aerosol concentrations at varying resolutions is associated more with the variation of the response of strong updrafts to aerosol-induced invigoration at varying resolutions than with that of weak updrafts. Another point to make here is that the frequency of weak updrafts is overestimated while that of strong updrafts is underestimated at coarse resolutions compared to the frequencies in the fine-resolution CSRM simulations.




### 5.1.4    Evaporation and precipitation distributions


Aerosol-induced increases in evaporation and associated cooling affect downdrafts, and changes in
downdrafts in turn affect gust fronts. Aerosol-induced changes in the intensity of gust fronts affect the
organization of cloud systems, which is characterized by cloud-cell spatiotemporal distributions. In
general, aerosol-induced greater increases in evaporation result in aerosol-induced greater changes in
the intensity of gust fronts and in cloud system organization (Tao et al., 2007, 2012; van den Heever
and Cotton, 2007; Storer et al., 2010; Lee et al., 2013, 2017).
Considering that individual cloud cells act as individual sources of precipitation, aerosol-induced
changes in the cloud system organization can alter precipitation spatiotemporal distributions, which
play an important role in hydrological circulations. It is thus important to examine how the response of
evaporation to increasing aerosol concentrations varies with varying resolution, i.e., to see how coarse
resolutions affect the quality of simulations of aerosol effects on hydrological circulations. Motivated
by this, evaporation rates are obtained and are shown in Figure 11.
As seen in the above-described variables, evaporation rates increase as the aerosol concentration
increases and the sensitivity of the evaporation rate to increasing aerosol concentrations reduces with
coarsening resolution among the ARW simulations. This suggests that the sensitivities of the cloud
system organization and precipitation distributions to increasing aerosol concentrations likely also
reduce with coarsening resolution, as reported in previous studies (e.g., Tao et al., 2007, 2012; van den



Heever and Cotton, 2007; Storer et al., 2010; Lee et al., 2013, 2017). This is confirmed by the
distribution of normalized precipitation frequency over precipitation rates shown in Figure 12. Similar
to the normalization for the updraft frequency, we first calculate the frequency of surface precipitation
rates at each time step and in each discretized precipitation rate bin. The frequency in each bin and at
each time step is then divided by the total number of grid points at the surface. The normalized
frequency at each time step is summed over all of the time steps. This sum is divided by the total
number of time steps as the final step in the normalization process. Figure 12 shows that due to the
reduction in the sensitivity of evaporative cooling to increasing the aerosol concentration as the
resolution coarsens, differences in the distribution of precipitation frequency between the polluted
scenario and the clean scenario reduce substantially as the resolution coarsens. Taking the 500-m
resolution CSRM simulations as benchmark simulations, this suggests that the coarse-resolution GFS
simulations likely underestimate the sensitivity of evaporative cooling, cloud system organization, and
precipitation distributions to increasing aerosol concentrations.

**5.2 Test on the effects of microphysics parameterizations on the simulations of clouds,**

**precipitation, and CAPI**


As mentioned previously, among microphysical processes, saturation, sedimentation, and collection
processes are those whose parameterizations are a main cause of errors in the simulation of clouds,
precipitation, and CAPI. Motivated by this, we focus on these three microphysical processes for testing



the effects of microphysics parameterizations on the simulations of clouds, precipitation, and CAPI.
As a preliminary step to this test, we first focus on the effects of microphysics parameterizations on the
simulation of the cloud mass, which plays a key role in cloud radiative properties and precipitation.
Based on Figures 3 and 4, we focus on the CLC, which accounts for the bulk of the total cloud mass.
Figure 13 shows the vertical distributions of the time- and domain-averaged CLC. In Figure 13a,
solid red and black lines represent the high-aerosol-Seoul run and the low-aerosol-Seoul run,
respectively, while in Figure 13b, those lines represent the high-aerosol-Houston run and the low-
aerosol-Houston run, respectively. Note that these runs shown in the figure are performed using the bin
scheme and the 500-m resolution. These simulations were repeated with the Morrison two-moment
scheme. These repeated simulations using the MG scheme, referred to as the high-aerosol-MG-Seoul
run, the low-aerosol-MG-Seoul run, the high-aerosol-MG-Houston run and the low-aerosol-MG-
Houston run, are represented by solid yellow and green lines in Figure 13. Between the high-aerosol
and low-aerosol runs using the MG scheme for the two cases, there is an increase in CLC with
increasing aerosol concentration. However, this increase is much smaller than that between the high-
aerosol and low-aerosol runs using the bin scheme for the two cases. In addition, there is a significant
difference in the shape of the vertical profile of CLC between the simulations with the MG scheme and
those with the bin scheme for both cases. Here, the shape is represented by the peak value of CLC and
the altitude of the peak value in the vertical profile. The peak value is higher in the simulations with the
bin scheme than in the simulations with the MG scheme for each of the polluted and clean scenarios.
The altitude of the peak value is lower in the simulations with the bin scheme than in the simulations



with the MG scheme. For the Seoul (Houston) case, the altitude is ~2 (3) km in the simulations with
the bin scheme, while it is ~5 km in those with the MG scheme.

We next test how the parameterization of saturation processes affects the simulations by

comparing the supersaturation prediction in the bin scheme to the saturation adjustment in the MG
scheme. To do this, the simulations with the bin scheme are repeated after replacing the supersaturation
prediction in the bin scheme with the saturation adjustment in the MG scheme. These repeated
simulations are referred to as the high-aerosol-sat-Seoul run, the low-aerosol-sat-Seoul run, the high-
aerosol-sat-Houston run, and the low-aerosol-sat-Houston run. The high-aerosol-sat-Seoul run and the
low-aerosol-sat-Seoul run for the Seoul case and the high-aerosol-sat-Houston run and the low-aerosol-
sat-Houston run for the Houston case are represented by dashed lines in Figure 13. As in the other
simulations, there is an increase in CLC with increasing aerosol concentrations between the high-
aerosol-sat and the low-aerosol-sat runs for the two cases. However, this increase is much smaller than
that between the high-aerosol and low-aerosol runs for the two cases, but is similar to that between the
high-aerosol-MG and low-aerosol-MG runs for the two cases.  This suggests that the sensitivity of the
CLC to increasing aerosol concentrations is affected by the parameterization of the saturation process
and that the use of the saturation adjustment reduces the sensitivity compared to using the
supersaturation prediction.

The high-aerosol-sat-Seoul run, the low-aerosol-sat-Seoul run, the high-aerosol-sat-Houston run,

and the low-aerosol-sat-Houston run are repeated by replacing the bin-scheme sedimentation with the
sedimentation from the MG scheme as a way of testing the effects of the parameterization of



sedimentation on the simulations. These repeated runs are referred to as the high-aerosol-sed-Seoul
run, the low-aerosol-sed-Seoul run, the high-aerosol-sed-Houston run, and the low-aerosol-sed-Houston
run. These runs are identical to the high-aerosol-Seoul run, the low-aerosol-Seoul run, the high-aerosol-
Houston run and the low-aerosol-Houston run, respectively, except for the parameterization of the
saturation and sedimentation processes. As mentioned previously, terminal velocities vary as
hydrometeor sizes vary in the bin scheme, while the MG scheme adopts mass-weight mean terminal
velocities for the calculation of the sedimentation process.

The vertical distributions of the CLC in the high-aerosol-sed-Seoul run, the low-aerosol-sed-Seoul

run, the high-aerosol-sed-Houston run, and the low-aerosol-sed-Houston run are represented by dashed
lines in Figure 14. Comparisons between the pair of high-aerosol-sed and low-aerosol-sed runs for the
two cases and the pair of high-aerosol-MG and low-aerosol-MG runs for the two cases show that not
only the increases in the CLC with increasing aerosol concentrations but also the shapes of the vertical
distribution of the CLC in the high-aerosol-sed and low-aerosol-sed runs for the two cases are similar to
those in the high-aerosol-MG and low-aerosol-MG runs for the two cases. This demonstrates that
differences in the shape of the vertical profile of CLC between the bin-scheme simulations and the MG-
scheme simulations are not explained by differences in the representation of the saturation process
alone. This also demonstrates that the representation of the sedimentation process plays an important
role in generating the differences in the shape of the vertical profile of CLC.

In Figure 14, we still see differences in the vertical profiles of CLC between the high-aerosol-sed-

Seoul and high-aerosol-MG-Seoul runs, and between the low-aerosol-sed-Seoul and low-aerosol-MG-





Seoul runs, as well as between the high-aerosol-sed-Houston and high-aerosol-MG-Houston runs, and between the low-aerosol-sed-Houston and low-aerosol-MG-Houston runs. To understand the cause of these differences, the high-aerosol-sed-Seoul run, the low-aerosol-sed-Seoul run, the high-aerosol-sed-Houston run, and the low-aerosol-sed-Houston run are repeated again with the MG-scheme collection process. These repeated runs are referred to as the high-aerosol-col-Seoul run, the low-aerosol-col-Seoul run, the high-aerosol-col-Houston run, and the low-aerosol-col-Houston run. These runs are identical to the high-aerosol-Seoul run, the low-aerosol run-Seoul, the high-aerosol-Houston run, and the low-aerosol-Houston run, respectively, except for the parameterization of the saturation, sedimentation, and collection processes. As mentioned previously, collection efficiencies vary as hydrometeor sizes vary in the bin scheme, while the MG scheme uses constant collection efficiencies.

As seen in Figure 15, the remaining differences between the high-aerosol-col-Seoul and high-aerosol-MG-Seoul runs and between the low-aerosol-col-Seoul and low-aerosol-MG-Seoul runs, as well as between the high-aerosol-col-Houston and high-aerosol-MG-Houston runs, and between the low-aerosol-col-Houston and low-aerosol-MG-Houston runs nearly disappear. This demonstrates with fairly good confidence that differences between the high-aerosol-Seoul run (the high-aerosol-Houston run) and the high-aerosol-MG-Seoul run (the high-aerosol-MG-Houston run) or between the low-aerosol-Seoul run (the low-aerosol-Houston run) and the low-aerosol-MG-Seoul run (the low-aerosol-MG-Houston run) are explained by differences in the parameterizations of the saturation, sedimentation, and collection processes between the bin scheme and the MG scheme.





### 5.3 Relative importance of resolution and parameterizations

Comparisons between ARW simulations with different resolutions and those with different microphysics parameterizations as shown in Figures 3 and 13 demonstrate that the variation in cloud variables is much greater with respect to the variation in resolution than with the variation in microphysics parameterizations. For example, comparisons between Figures 3 and 13 show that the variation in the time- and domain-averaged cloud mass is ~2–4 times greater as the resolution varies than when the microphysics parameterizations varies. These comparisons also show that the variation in cloud variables with varying resolutions explains the discrepancy between GFS simulations and CSRM simulations and between GFS simulations and observations much better than the variation in microphysics parameterizations. As a first step toward reducing the first-order errors in the GFS simulations, we first need to focus on the reduction in errors that are associated with the use of coarse resolutions in the GFS model.

### 6. Summary and Discussion

This study examines the uncertainties in the simulations of clouds, precipitation, and CAPI in the NWP models. Here, we focus on those uncertainties that are created by the microphysics parameterizations and by the model resolution chosen. In particular, for the examination of the uncertainties associated



with microphysics parameterizations, we investigate the contributions of the parameterizations of
three key microphysical processes, i.e., saturation, collection, and sedimentation, to the uncertainties.

As a way of examining the uncertainties created by the microphysics parameterizations, we

compare the MG scheme (a representative bulk scheme) to the bin scheme, which acts as a benchmark
scheme. The vertical distribution of the cloud mass simulated by the MG scheme deviates substantially
from that simulated by the bin scheme. In particular, there is a substantial discrepancy in the peak value
of the distribution and the altitude of the peak value between the schemes. Also, there is a substantial
discrepancy between the schemes in the sensitivity of the cloud mass to increasing aerosol
concentrations.

The discrepancy in the sensitivity is closely linked to the discrepancy in the parameterization of the

saturation processes between the schemes. The use of the saturation adjustment in the bulk scheme
reduces the sensitivity by a factor of ~2 compared to the use of the supersaturation prediction in the bin
scheme. The discrepancy in the peak value and its altitude between the schemes is strongly linked to the
parameterization of sedimentation in the schemes. The use of identical parameterizations of saturation
and sedimentation makes the sensitivity and the peak value and its altitude similar between the schemes,
although there still remains a slight difference in the magnitude of the cloud mass. This remaining
difference is explained by the discrepancy in the parameterization of the collection process. When the
two schemes use identical parameterizations of the saturation, sedimentation, and collection processes,
the sensitivity and the peak value and its altitude become nearly identical between the two schemes.
This confirms that differences in the parameterizations of the three key processes (i.e., saturation,



sedimentation, and collection) are the main cause of the differences in the simulations of clouds between the schemes as indicated by Fan et al. (2012) and Khain et al. (2015).

By selecting the simulations with the bin scheme as benchmark simulations, we see that the use of the saturation adjustment, as done in most current NWP models, can lead to an underestimation of the sensitivity of the cloud mass to increasing aerosol concentrations. Fan et al. (2012) and Khain et al. (2015) have also shown that the sensitivity of the cloud mass to increasing aerosol concentrations is lower in the bulk scheme than in the bin scheme. This study shows that the lower sensitivity in the bulk scheme is closely linked to the use of the saturation adjustment in the bulk scheme.

It is well known that the shape of the vertical profile of the cloud mass (i.e., the peak value of the cloud mass and its altitude) or how cloud mass is distributed in the vertical domain has substantial implications for cloud radiative forcing and precipitation processes. This study demonstrates that the different parameterizations of the sedimentation process between the schemes lead to different shapes of the cloud-mass profiles and thus different cloud radiative forcings and precipitation processes. The use of a mass-weight mean terminal velocity for sedimentation as done in the bulk schemes can lead to misleading shapes, cloud radiative forcings, and precipitation processes compared to those in the benchmark bin-scheme simulations where terminal velocities vary as hydrometeor sizes vary.

NWP models (e.g., the GFS model) adopt coarse resolutions. This study shows that the use of coarse resolutions can cause an underestimation of the updraft intensity and thus supersaturation, which leads to an underestimation of the cloud mass. Also, the use of coarse resolutions likely results in the





underestimation of the sensitivity of updrafts and cloud mass and that of evaporation, cloud system
organization, and precipitation distributions to increasing aerosol concentrations.
Through the examination of the sensitivity of the results to the resolution chosen, we find that
updrafts, associated other cloud variables, and their sensitivity to increasing aerosol concentrations are
strongly controlled by small-scale updrafts. When they are resolved with the use of high-resolution
models, there are high-level averaged updrafts, associated variables, and their sensitivity but when they
are not resolved in low-resolution models, there are low-level averaged updrafts, associated variables,
and their sensitivity. This means that small-scale updrafts not resolved with coarse resolutions play an
important role in the simulation of the correct magnitude of updrafts, associated variables, and their
sensitivity to increasing aerosol concentrations.
The frequency distributions of updrafts simulated in this study show that the frequency of weak
updrafts is overestimated while that of strong updrafts is underestimated in the simulations with coarse
resolutions compared to those in the CSRM simulations. Hence, the updraft speed shifts toward lower
values with coarsening resolution. The difference in the frequency between the polluted and clean
scenarios reduces substantially, particularly for strong updrafts, with coarsening resolution. This is why
the sensitivity of updrafts and associated cloud variables to increasing aerosol concentrations reduces
with coarsening resolution. We see that not resolving small-scale updrafts results in the underestimation
of strong updrafts and the overestimation of weak updrafts for both scenarios and in the reduced
difference in strong updrafts between the scenarios.



The GFS simulations use the so-called sub-grid parameterizations (e.g., cumulus
parameterizations) that represent sub-grid updrafts and associated variables, while the ARW
simulations at the 500-m resolution (i.e., the CSRM simulations) do not use these sub-grid
parameterizations based on consideration that the CSRM simulations resolve sub-grid processes. Thus,
the CSRM simulations (that prove to act as benchmark simulations through comparisons to
observations) are able to evaluate the sub-grid parameterizations in the GFS model. The sub-grid
parameterizations are designed to correct errors that are caused by the use of coarse resolutions in the
GFS model. However, comparisons between the GFS simulations and the ARW simulations at different
resolutions indicate that despite the presence of sub-grid parameterizations in the GFS model, the errors
or differences in the updraft intensity and associated cloud variables between the GFS simulations and
the CSRM simulations exist due to resolutions. Hence, sub-grid parameterizations need to be improved
to better represent sub-grid processes. To this end, results here indicate that sub-grid parameterizations
(e.g., scale-aware cumulus schemes) which are being implemented into the NWP models (e.g., the GFS
model) should be able to compensate for the over- and under-estimation of weak updrafts and strong
updrafts, respectively, due to coarse resolutions.
Comparisons between GFS simulations and ARW simulations also indicate that it is likely that
the GFS model underestimates the sensitivity of updrafts and associated cloud variables to increasing
aerosol concentrations. In general, parameterizations that represent sub-grid updrafts and other
associated variables do not have pathways through which increasing aerosol concentrations affect
updrafts and associated cloud variables.  However, recent studies by Lim et al. (2014), Thayer-Calder et



al. (2015), and Griffin and Larson (2016) have attempted to consider interactions among microphysical processes, their variations with varying aerosol concentrations, and sub-grid dynamic (e.g., updrafts and downdrafts) and thermodynamic (e.g., temperature) variables in those parameterizations. These efforts should focus on countering the variation in the sensitivity of updrafts, in particular strong updrafts and thus that of cloud variables, cloud system organization, and precipitation distributions to increasing aerosol concentrations with coarsening resolution. While the pattern of the sensitivity and its variation shown in this study provides valuable information useful for aiding these efforts, results may be different for different cloud types and environments, given the strong dependence of aerosol-cloud interactions on cloud type and environmental conditions. So to aid the efforts in a generalized way, future studies with more cases that involve various types of aerosol-cloud interactions are needed.





**Acknowledgements.** This study is supported by NOAA (Grant NOAA-NWS-NWSPO-2015-
2004117), which also provided the GFS forecast data.


























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



**Tables**

Table 1. Description of the simulations.

| Simulations | Case | Aerosol number concentration at the surface (cm$^{-3}$) | Microphysics scheme | Resolution | Saturation | Sedimentation | Collection |
|---|---|---|---|---|---|---|---|
| High-aerosol-Seoul run | Seoul | 5500 | Bin | 500 m | Supersaturation prediction | Bin-scheme sedimentation | Bin-scheme collection |
| Low-aerosol-Seoul run | Seoul | 550 | Bin | 500 m | Supersaturation prediction | Bin-scheme sedimentation | Bin-scheme collection |
| High-aerosol-Houston run | Houston | 1500 | Bin | 500 m | Supersaturation prediction | Bin-scheme sedimentation | Bin-scheme collection |
| Low-aerosol-Houston run | Houston | 150 | Bin | 500 m | Supersaturation prediction | Bin-scheme sedimentation | Bin-scheme collection |
| High-aerosol-15-Seoul run | Seoul | 5500 | Bin | 15 km | Supersaturation prediction | Bin-scheme sedimentation | Bin-scheme collection |
| Low-aerosol-15-Seoul run | Seoul | 550 | Bin | 15 km | Supersaturation prediction | Bin-scheme sedimentation | Bin-scheme collection |
| High-aerosol-15-Houston run | Houston | 1500 | Bin | 15 km | Supersaturation prediction | Bin-scheme sedimentation | Bin-scheme collection |
| Low-aerosol-15-Houston | Houston | 150 | Bin | 15 km | Supersaturation prediction | Bin-scheme sedimentation | Bin-scheme collection |



| run | | | | | | | |
|---|---|---|---|---|---|---|---|
| High-areosol-35-Seoul run | Seoul | 5500 | Bin | 35 km | Supersaturation prediction | Bin-scheme sedimentation | Bin-scheme collection |
| Low-aerosol-35-Seoul run | Seoul | 550 | Bin | 35 km | Supersaturation prediction | Bin-scheme sedimentation | Bin-scheme collection |
| High-aerosol-35-Houston run | Houston | 1500 | Bin | 35 km | Supersaturation prediction | Bin-scheme sedimentation | Bin-scheme collection |
| Low-aerosol-35-Houston run | Houston | 150 | Bin | 35 km | Supersaturation prediction | MG-scheme sedimentation | MG-scheme collection |
| High-aerosol-MG-Seoul run | Seoul | 5500 | MG | 500 m | Saturation adjustment | MG-scheme sedimentation | MG-scheme collection |
| Low-aerosol-MG-Seoul run | Seoul | 550 | MG | 500 m | Saturation adjustment | MG-scheme sedimentation | MG-scheme collection |
| High-aerosol-MG-Houston run | Houston | 1500 | MG | 500 m | Saturation adjustment | MG-scheme sedimentation | MG-scheme collection |
| Low-aerosol-MG-Houston run | Houston | 150 | MG | 500 m | Saturation adjustment | MG-scheme sedimentation | MG-scheme collection |
| High-aerosol-sat-Seoul run | Seoul | 5500 | Bin | 500 m | Saturation adjustment | Bin-scheme sedimentation | Bin-scheme collection |
| Low-aerosol-sat-Seoul run | Seoul | 550 | Bin | 500 m | Saturation adjustment | Bin-scheme sedimentation | Bin-scheme collection |
| High-aerosol- | Houston | 1500 | Bin | 500 m | Saturation | Bin-scheme | Bin-scheme |





| sat-Houston run | | | | | adjustment | sedimentation | collection |
|---|---|---|---|---|---|---|---|
| Low-aerosol-sat-Houston run | Houston | 150 | Bin | 500 m | Saturation adjustment | Bin-scheme sedimentation | Bin-scheme collection |
| High-aerosol-sed-Seoul run | Seoul | 5500 | Bin | 500 m | Saturation adjustment | MG-scheme sedimentation | Bin-scheme collection |
| Low-aerosol-sed-Seoul run | Seoul | 550 | Bin | 500 m | Saturation adjustment | MG-scheme sedimentation | Bin-scheme collection |
| High-aerosol-sed-Houston run | Houston | 1500 | Bin | 500 m | Saturation adjustment | MG-scheme sedimentation | Bin-scheme collection |
| Low-aerosol-sed-Houston run | Houston | 150 | Bin | 500 m | Saturation adjustment | MG-scheme sedimentation | Bin-scheme collection |
| High-aerosol-col-Seoul run | Seoul | 5500 | Bin | 500 m | Saturation adjustment | MG-scheme sedimentation | MG-scheme collection |
| Low-aerosol-col-Seoul run | Seoul | 550 | Bin | 500 m | Saturation adjustment | MG-scheme sedimentation | MG-scheme collection |
| High-aerosol-col-Houston run | Houston | 1500 | Bin | 500 m | Saturation adjustment | MG-scheme sedimentation | MG-scheme collection |
| Low-aerosol-col-Houston run | Houston | 150 | Bin | 500 m | Saturation adjustment | MG-scheme sedimentation | MG-scheme collection |


**FIGURE CAPTIONS**

Figure 1. (a) Sea-level pressure (hPa) and (b) 850 hPa wind (m $s^{-1}$; arrows), geopotential height (m; contours) and equivalent potential temperature (K; shaded) at 0900 LST July $26^{th}$ 2011 over the northeast Asia. The rectangles in the Korean Peninsula in panels (a) and (b) mark the center of Seoul. (c) Sea-level pressure (hPa;shaded) and wind at 10 m above sea level (m $s^{-1}$; barbs) and (d) convective available potential energy (J $kg^{-1}$) at 0500 LST 18 July 2013 in and around Houston. The rectangles in panels (c) and (d) mark the center of Houston.

Figure 2. (a) The domain (marked by the rectangle) used in simulations for the Seoul case. The small white circle marks the center of Seoul. (b) The domain used in simulations for the Houston case. The small white circle marks the center of Houston.

Figure 3. Vertical distributions of the time- and domain-averaged cloud liquid content (CLC) for (a) the Seoul case and (b) the Houston case. Solid lines represent simulations at the 500-m resolution, while dashed lines represent those at the 15-km resolution. Dotted lines represent simulations at the 35-km resolution and blue lines represent GFS-simulated CLC.

Figure 4. Same as Figure 3, but for cloud ice content (CIC).





Figure 5. Time series of the domain-averaged (a) liquid water path (LWP) and (b) ice water path
(IWP) for the Seoul case. Solid lines represent simulations at the 500-m resolution, while dashed and
dotted lines represent those at 15-km and 35-km resolutions, respectively. Blue lines represent GFS-
simulated LWP and IWP and green lines represent observed LWP and IWP.

Figure 6. Same as Figure 5, but for the Houston case.

Figure 7. Vertical distributions of the time- and domain-averaged updraft mass fluxes for (a) the Seoul
case and (b) the Houston case. Solid lines represent simulations at the 500-m resolution, while dashed
lines represent those at the 15-km resolution. Dotted lines represent simulations at the 35-km resolution
and blue lines represent GFS-simulated updraft mass fluxes.

Figure 8. Vertical distributions of the time- and domain-averaged condensation rates for (a) the Seoul
case and (b) the Houston case. Solid lines represent simulations at the 500-m resolution, while dashed
lines represent those at the 15-km resolution. Dotted lines represent simulations at the 35-km resolution.

Figure 9. Same as Figure 8, but for deposition rates.





Figure 10. Distributions of normalized updraft frequency over updraft speeds for (a) the Seoul case

and (b) the Houston case. Solid lines represent simulations at the 500-m resolution, while dashed lines

represent those at the 15-km resolution. Dotted lines represent simulations at the 35-km resolution.

Figure 11. Same as Figure 8, but for evaporation rates.

Figure 12. Distributions of normalized precipitation frequency over precipitation rates for (a) the Seoul

case and (b) the Houston case. Solid lines represent simulations at the 500-m resolution, while dashed

lines represent those at the 15-km resolution. Dotted lines represent simulations at the 35-km resolution.

Figure 13. Vertical distributions of the time- and domain-averaged cloud liquid content (CLC) for (a)

the Seoul case and (b) the Houston case. Solid red and black lines represent simulations with the bin

scheme and at the 500-m resolution, while dashed red and black lines represent the bin-scheme

simulations with the saturation adjustment. Solid yellow and green lines represent simulations with the

MG scheme.

Figure 14. Vertical distributions of the time- and domain-averaged cloud liquid content (CLC) for (a)

the Seoul case and (b) the Houston case. Solid red and black lines represent simulations with the bin

scheme and at the 500-m resolution, while dashed red and black lines represent the bin-scheme





simulations with the saturation adjustment and the MG scheme sedimentation process. Solid yellow
and green lines represent simulations with the MG scheme.

Figure 15. Vertical distributions of the time- and domain-averaged cloud liquid content (CLC) for (a)
the Seoul case and (b) the Houston case. Solid red and black lines represent simulations with the bin
scheme and at the 500-m resolution, while dashed red and black lines represent the bin-scheme
simulations with the saturation adjustment and the MG scheme sedimentation and collection processes.
Solid yellow and green lines represent simulations with the MG scheme.














a

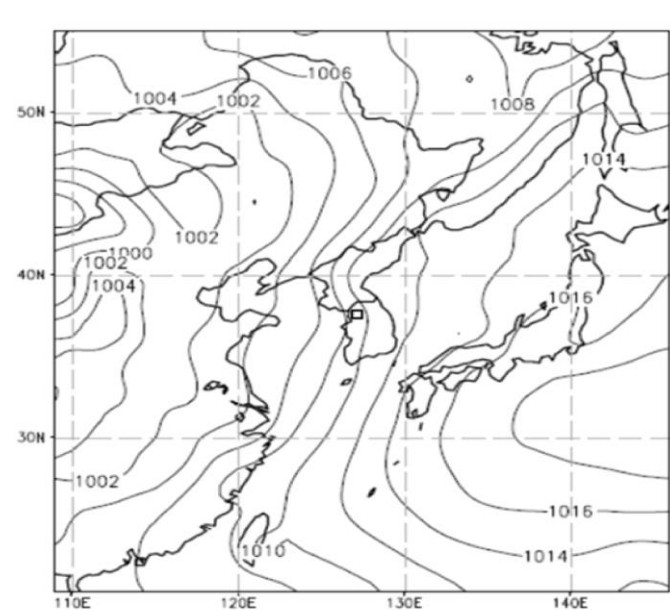

b

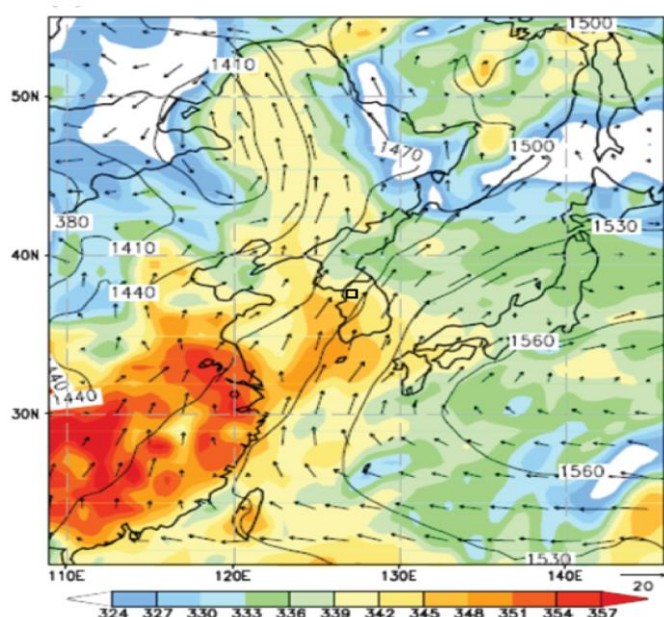


**Figure 1**





c

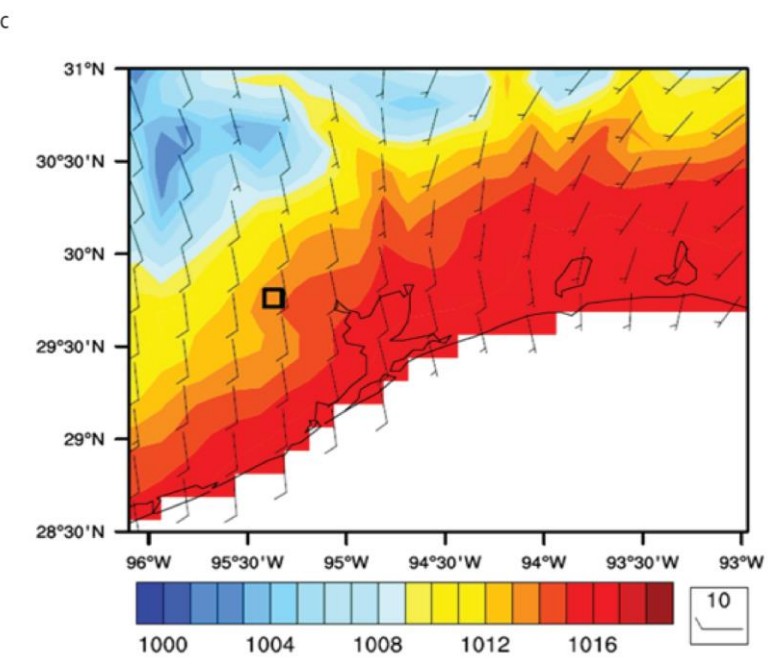

d

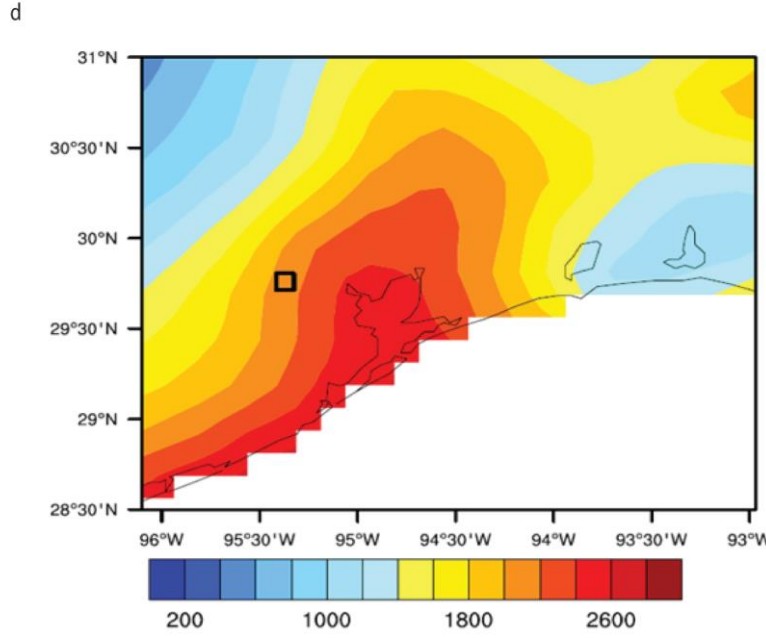



**Figure 1**



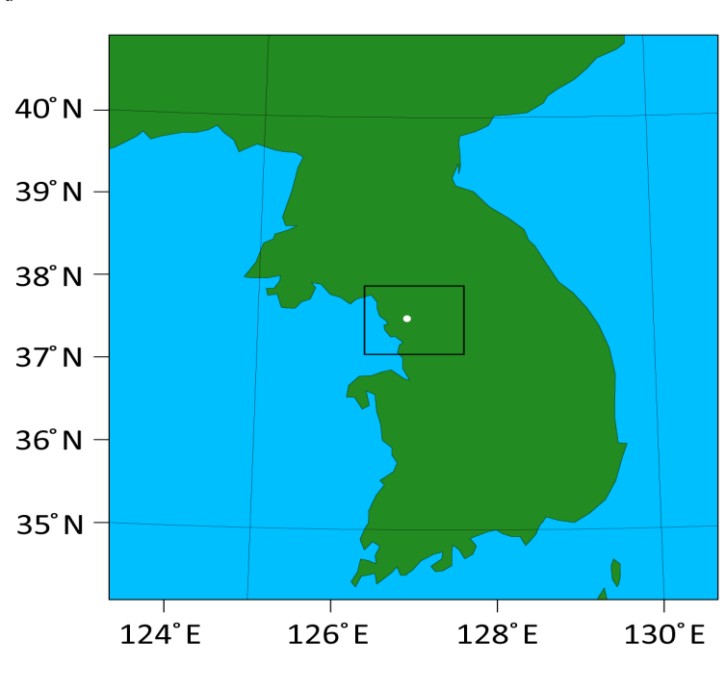

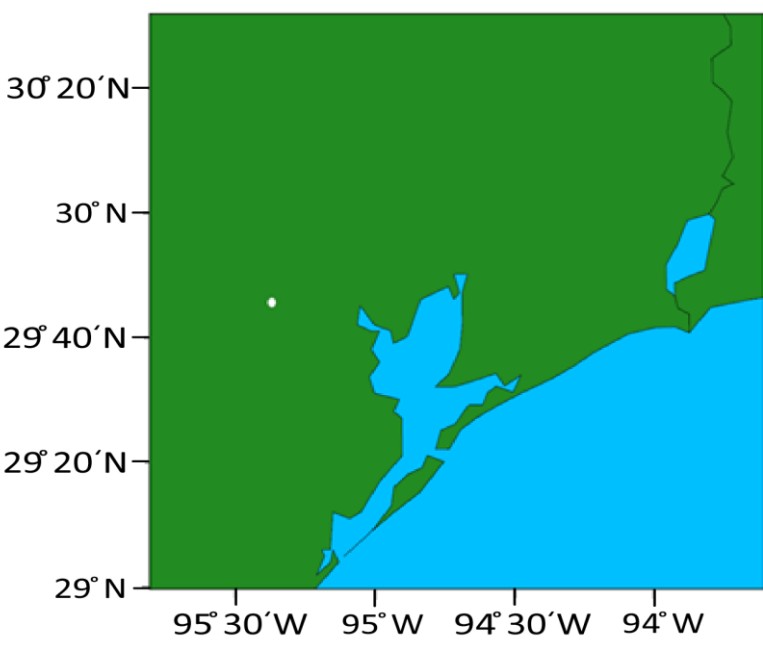

**Figure 2**




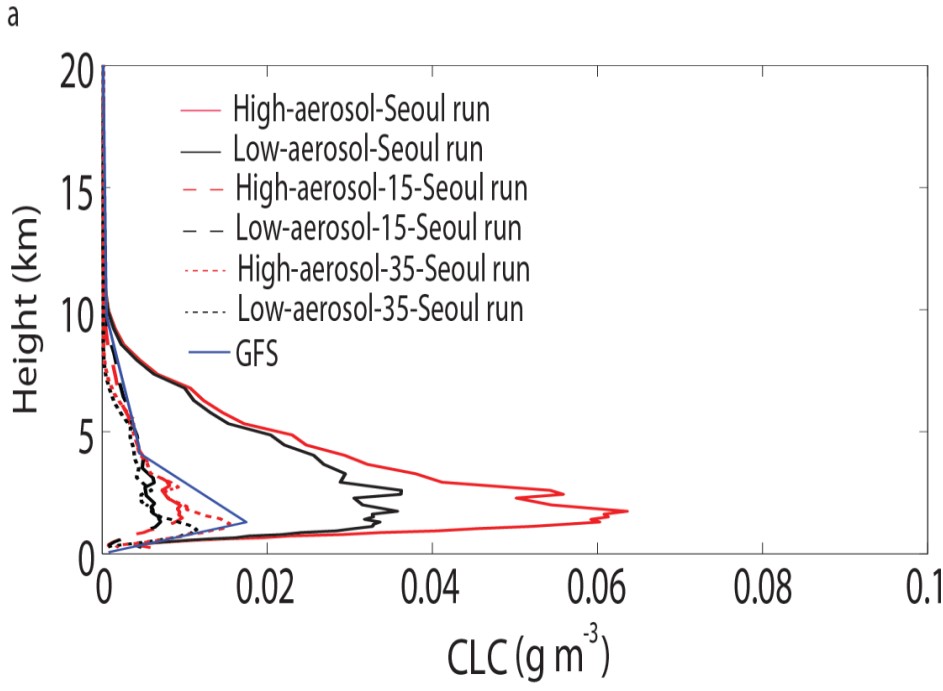

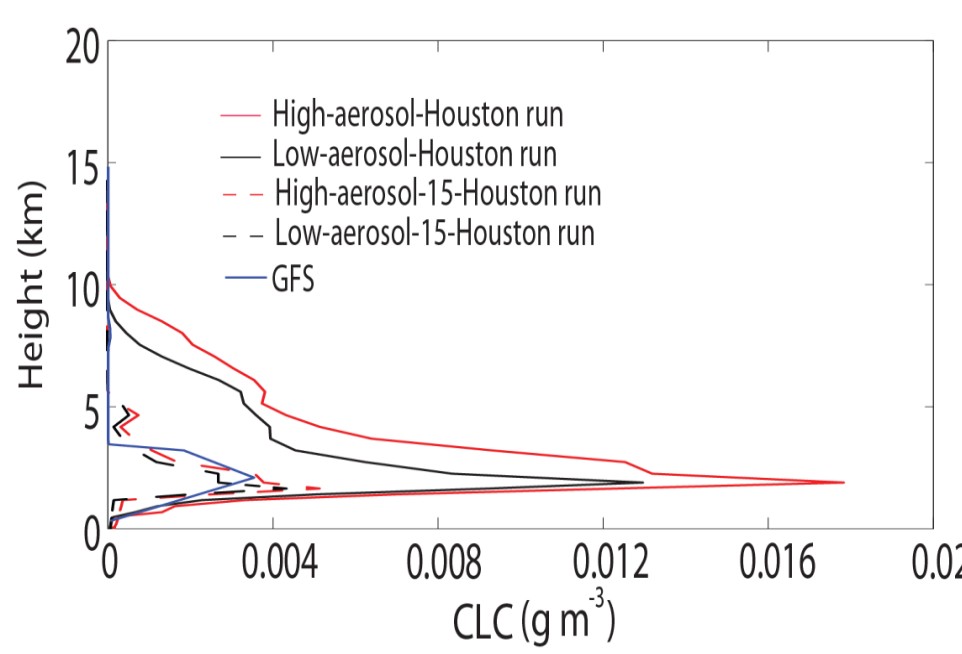


808                              **Figure 3**



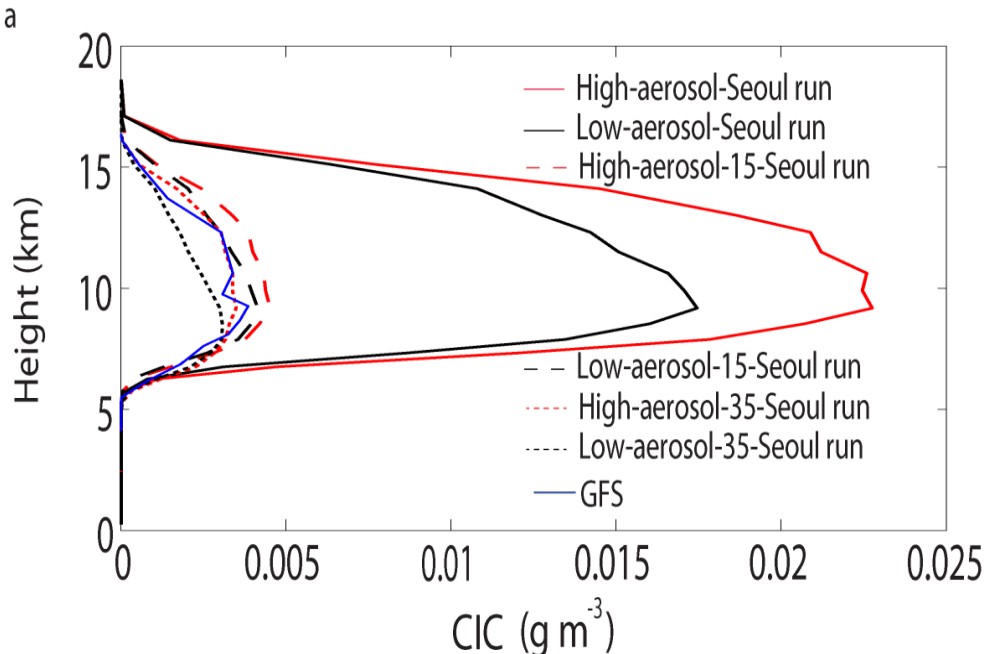

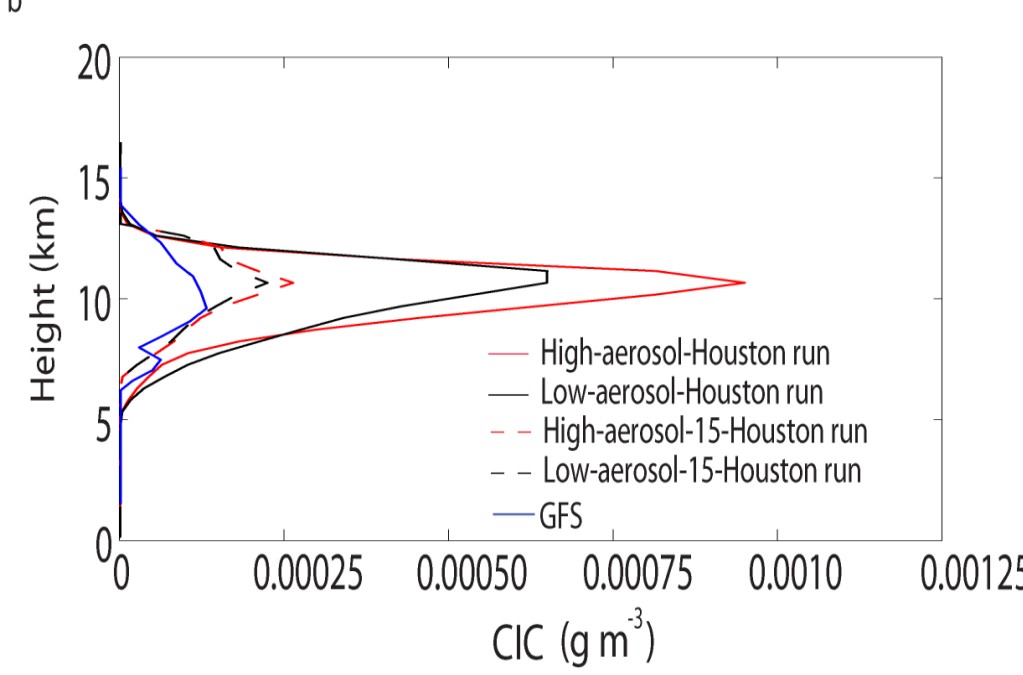


810                         **Figure 4**



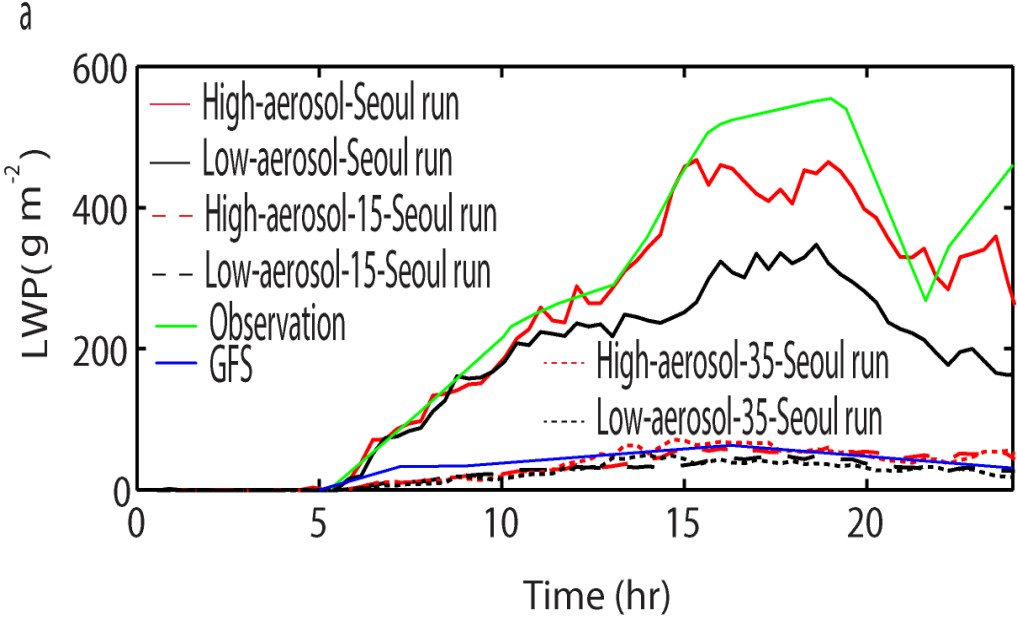

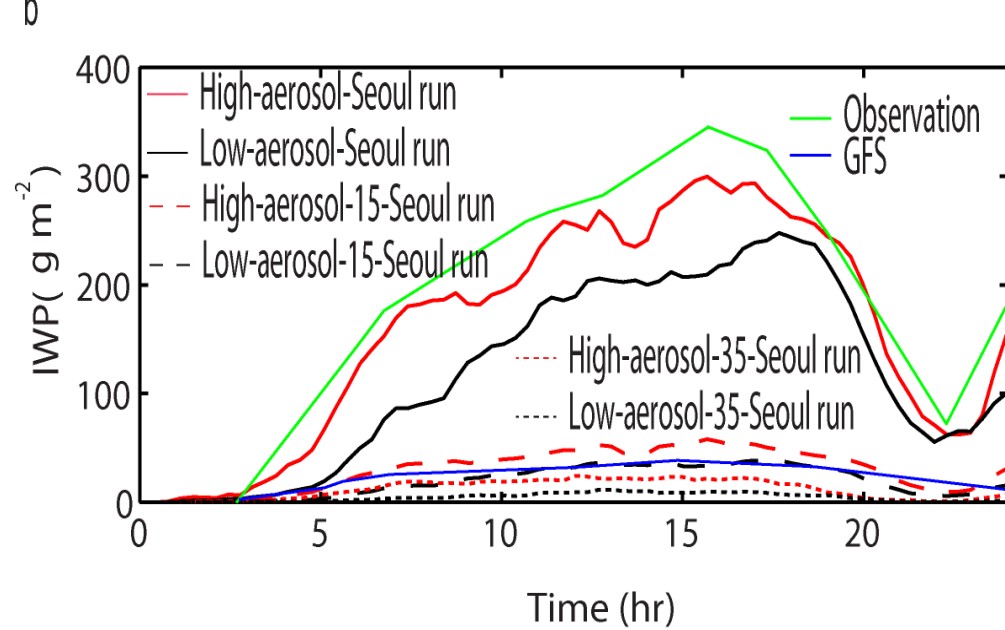


**Figure 5**





a

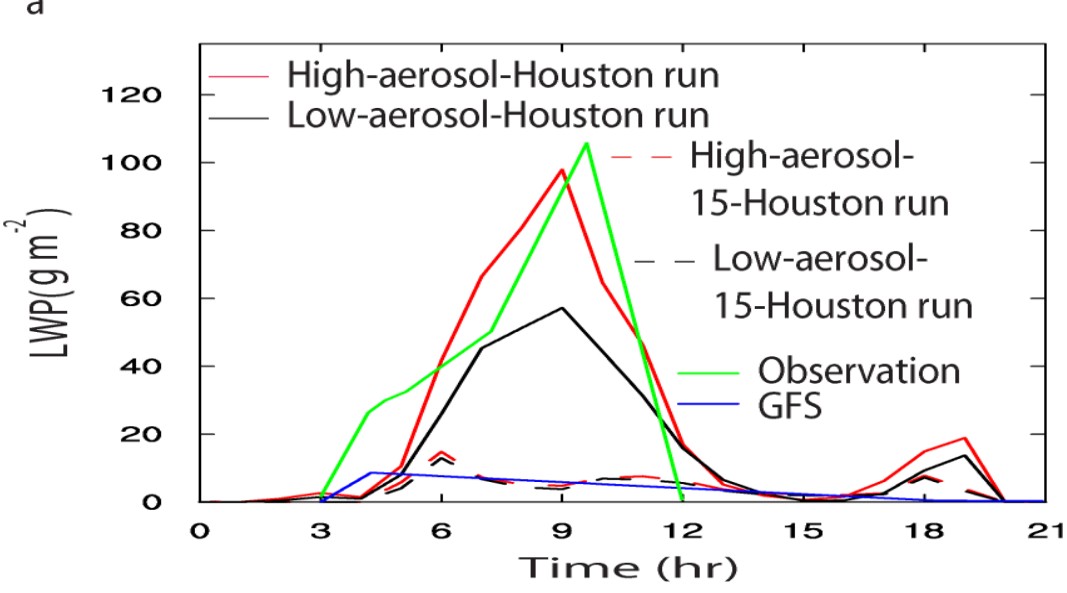

b

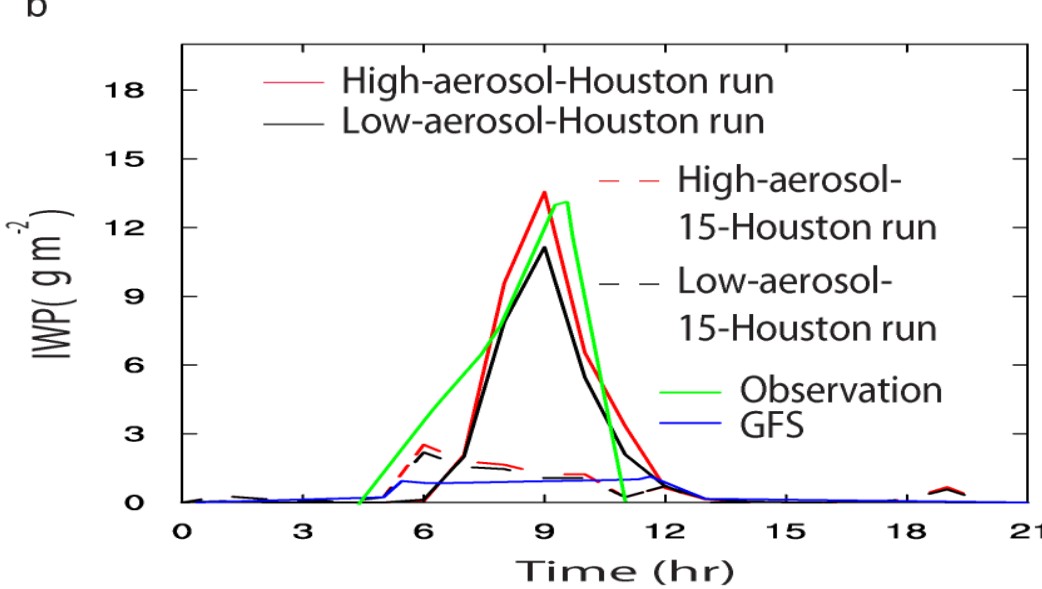


**Figure 6**



a

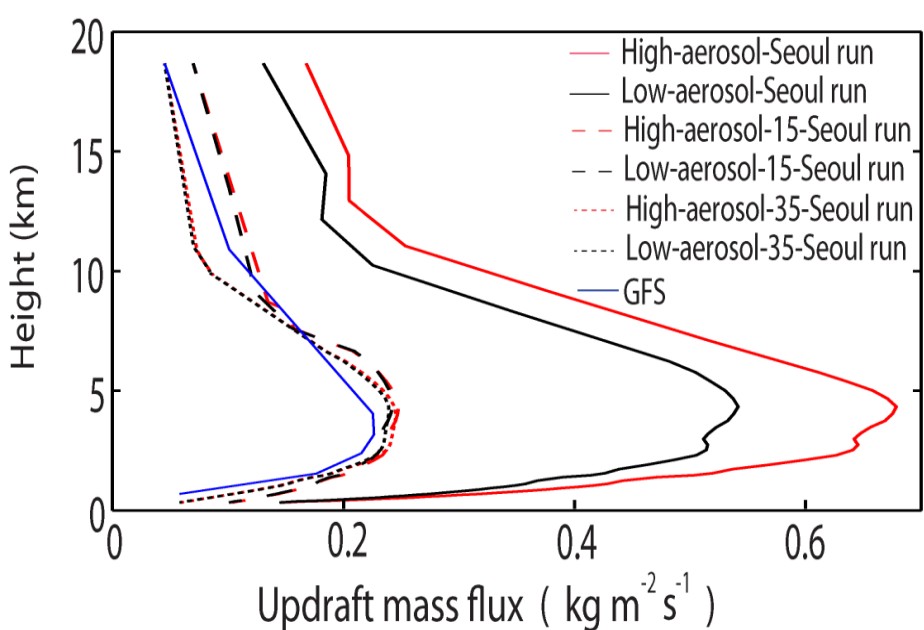

b

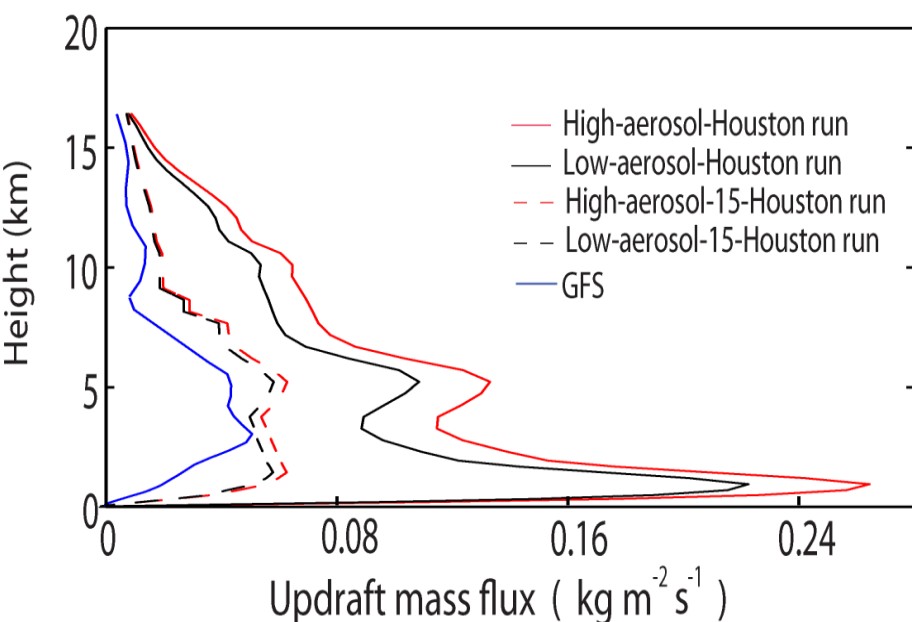


**Figure 7**




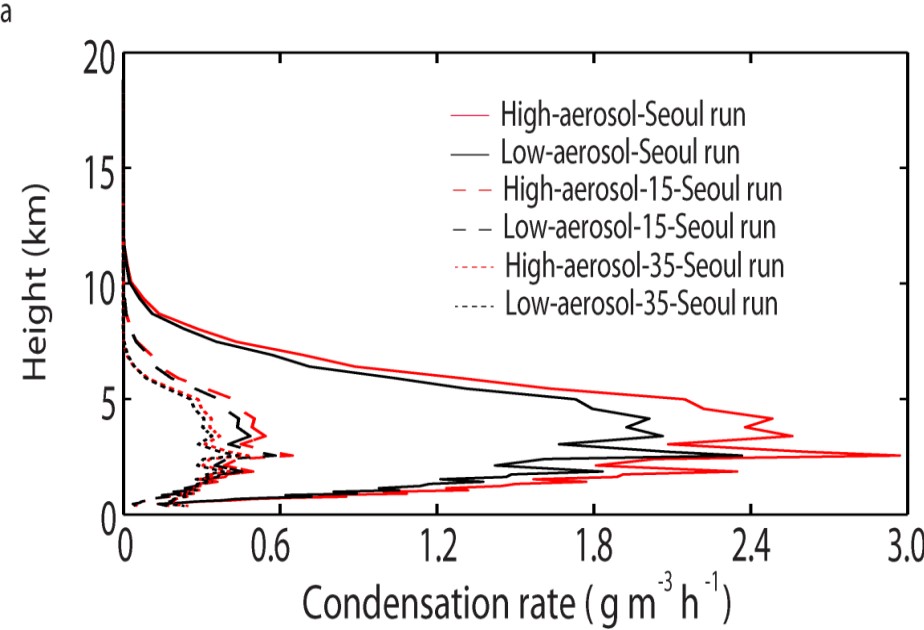

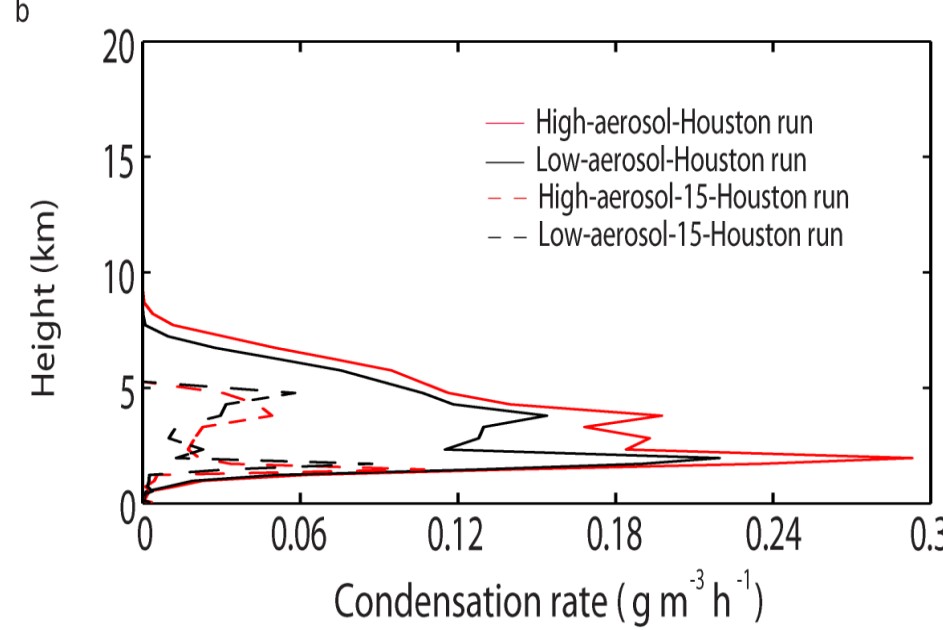


818                        **Figure 8**



a

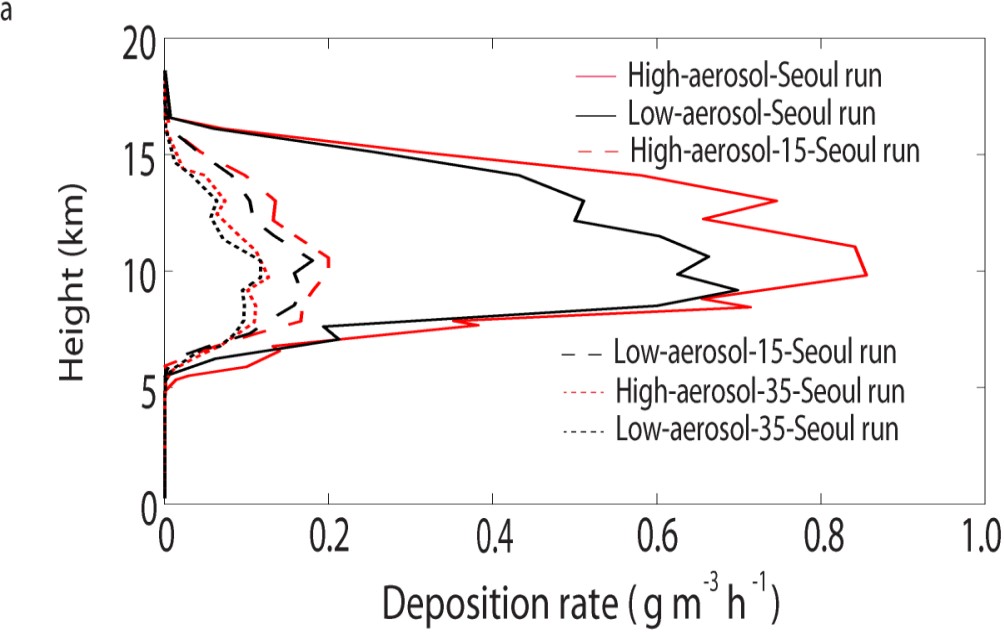

b

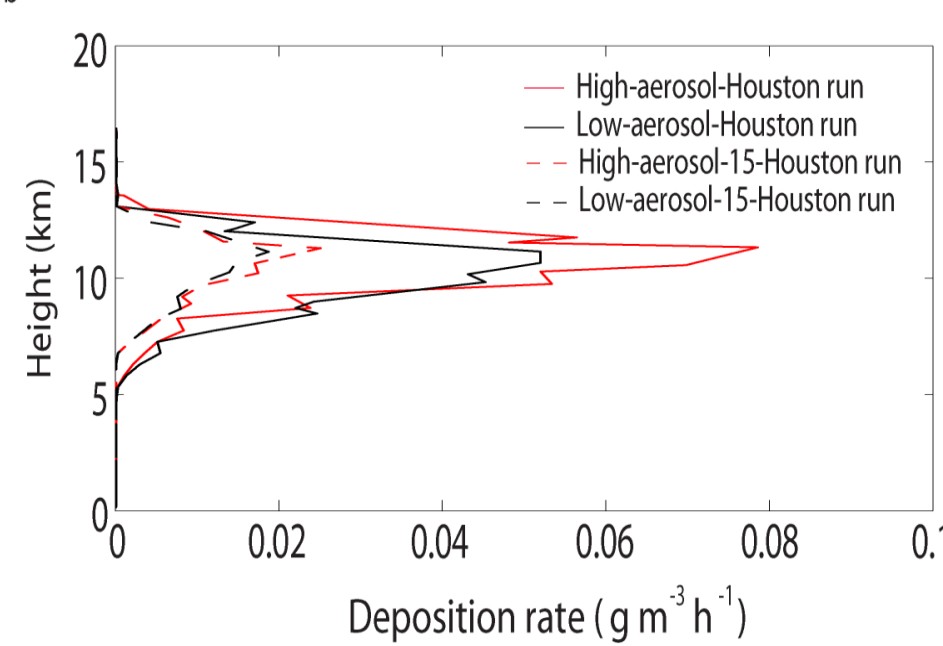


**Figure 9**



a

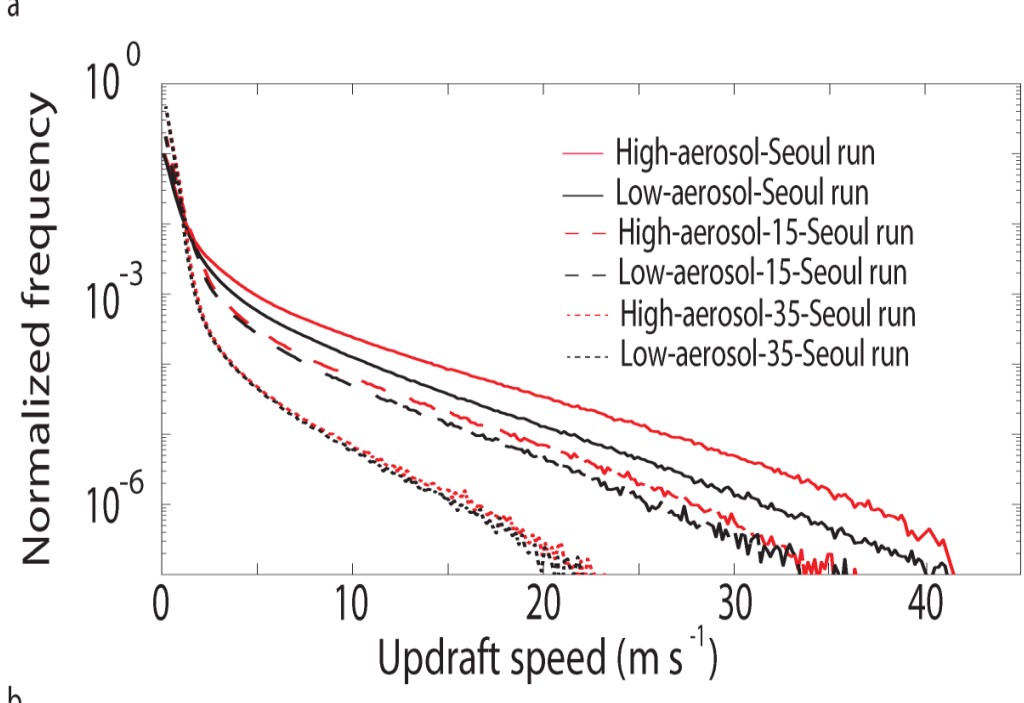

b

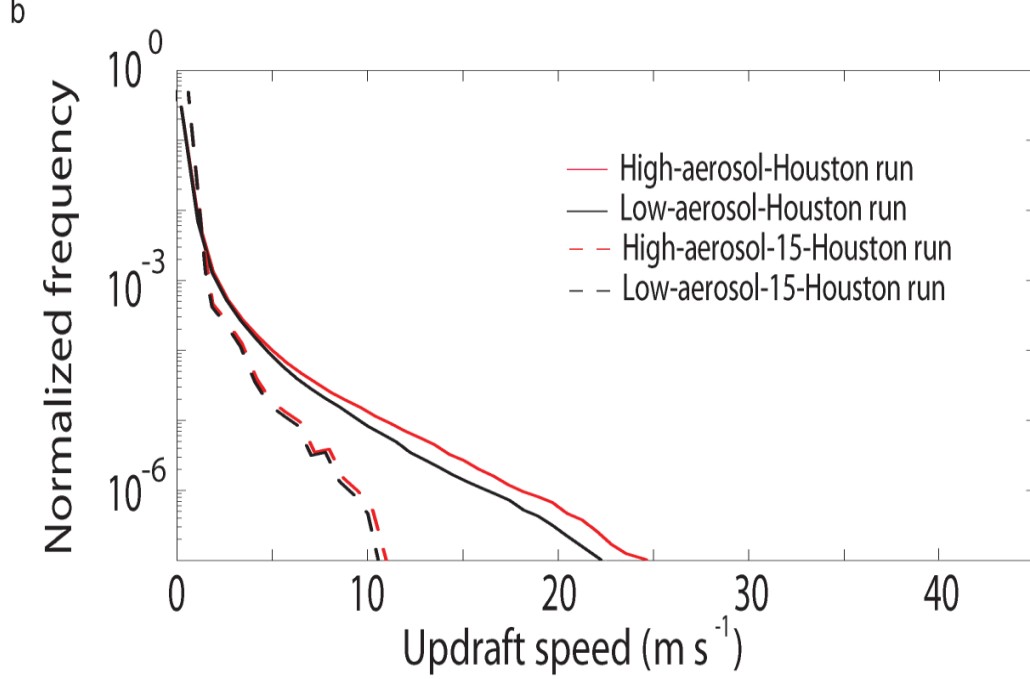


**Figure 10**



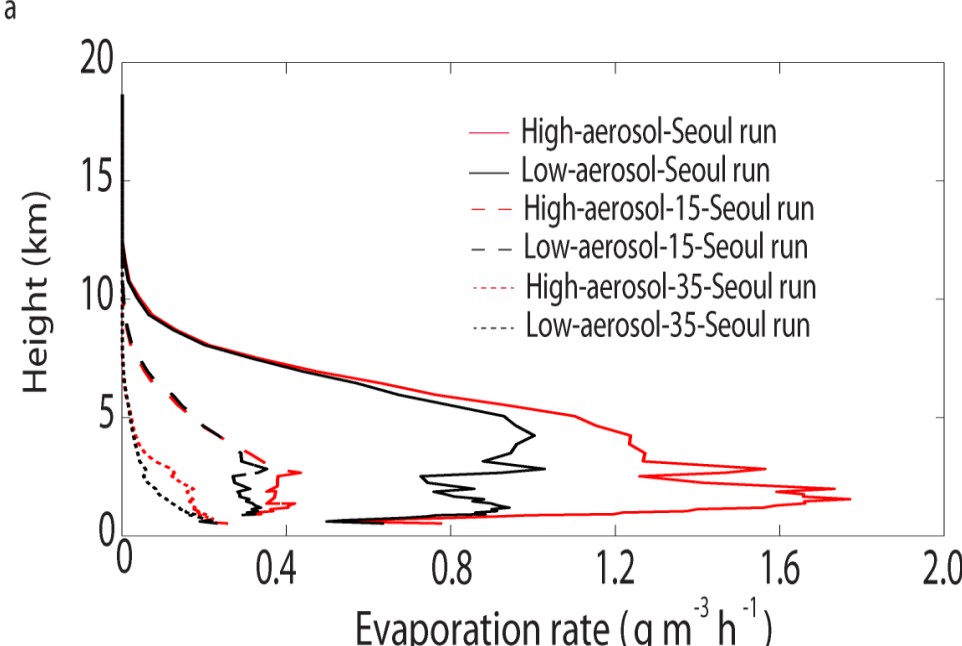

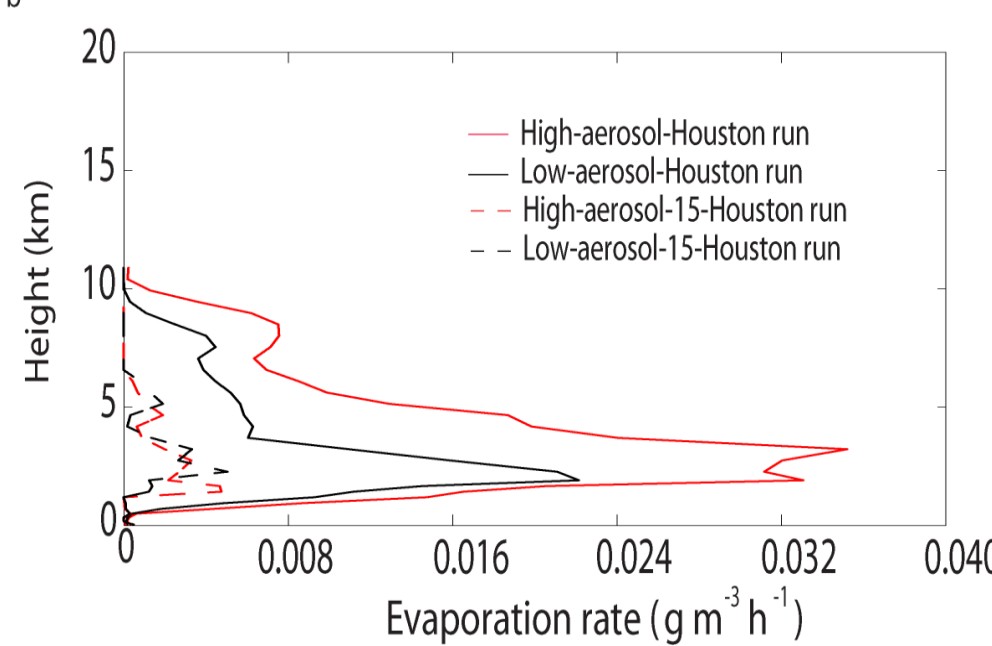


824                               **Figure 11**



a

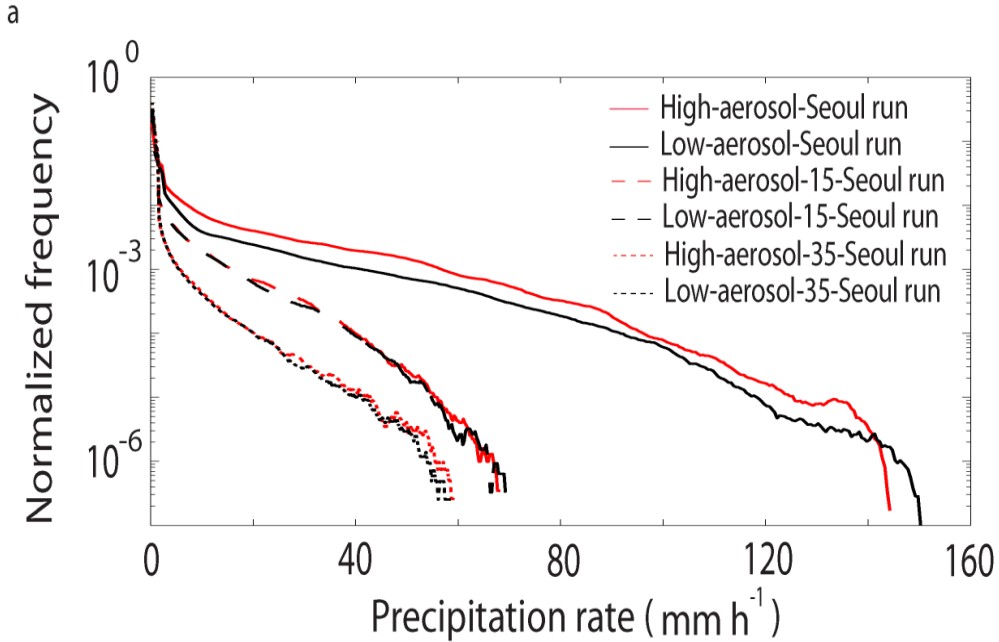

b

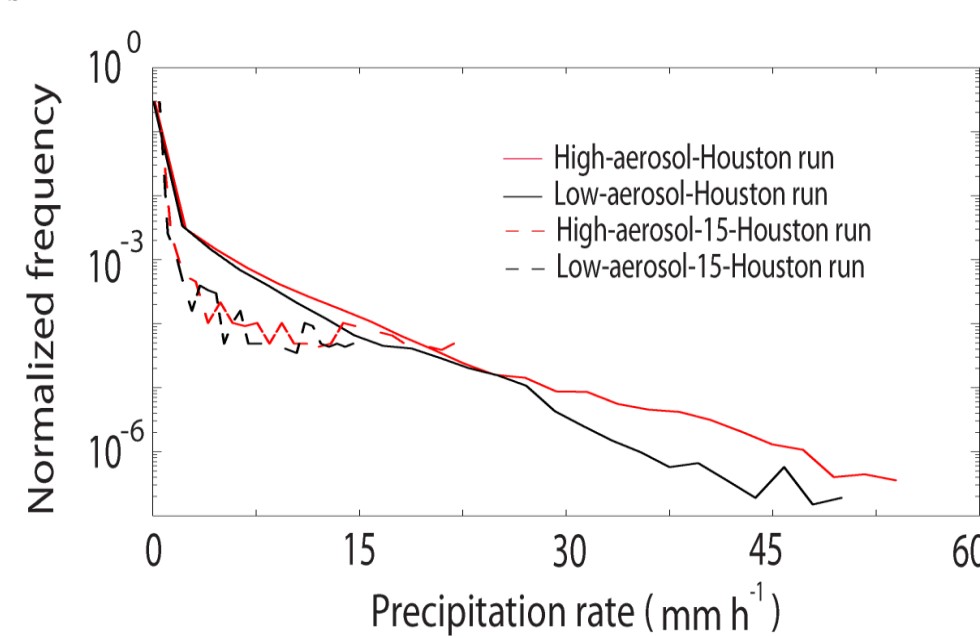

826                          **Figure 12**

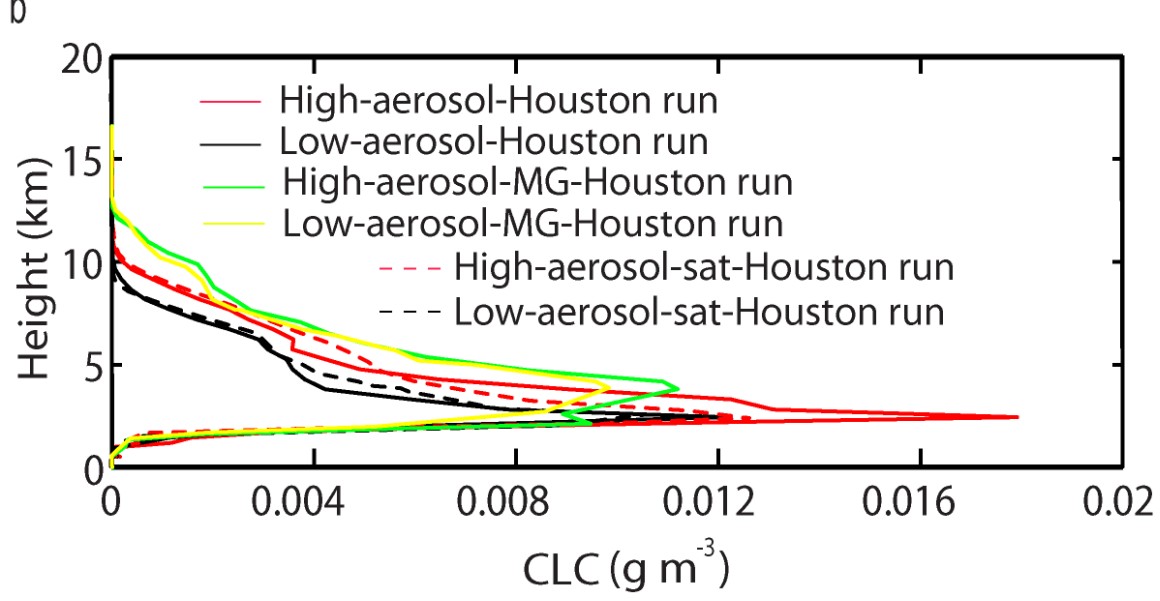


**Figure 13**





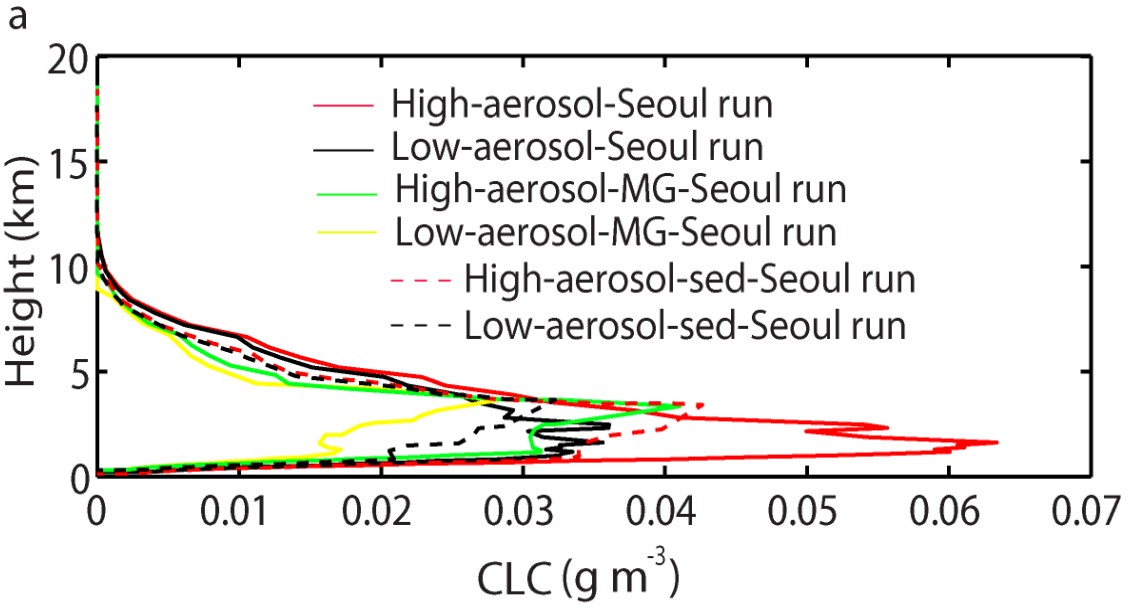

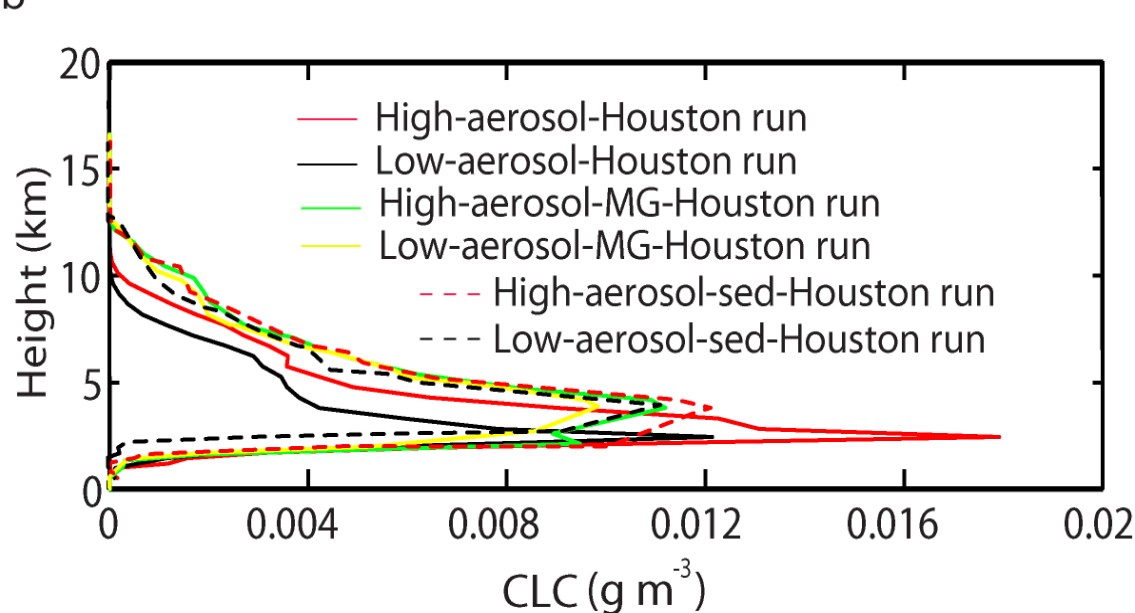

**Figure 14**





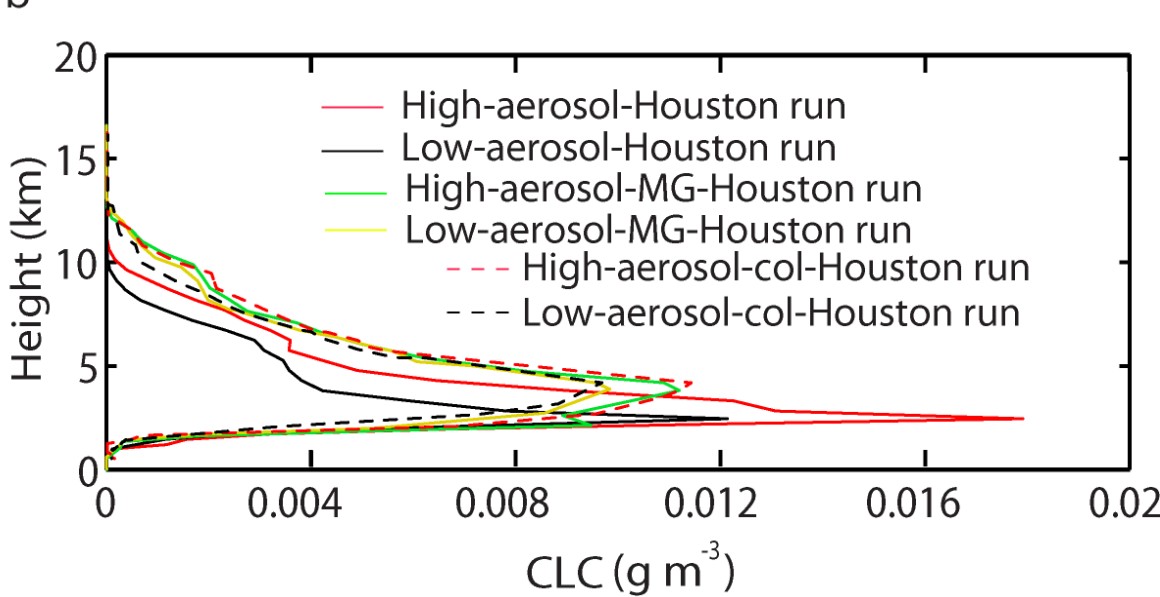


**Figure 15**

