# Peer review of "Effects of model resolution and parameterizations on the simulations of clouds, precipitation, and their interactions with aerosols"

_Atmospheric Chemistry and Physics, 2017_

## Referee Comment (RC1) · Anonymous Referee #1 · 22 Jun 2017

This manuscript is a bit of a mixed bag. I really like the analysis of the difference between the bin and bulk microphysics. The analysis of resolution dependence of the clouds and cloud-aerosol interactions is not so clear, as too little information is provided regarding the representation of deep convection at coarse resolution.

The illustrations are generally helpful, and the writing is mostly quite clear.

Lines 35-38. Is the comparison done at the same scale? We certainly wouldn't expect a coarse resolution simulation to produce the same updraft intensity as a fine resolution

simulation if they aren't compared at the same scale. So I'm withholding judgement on this conclusion until I know more. Perhaps need to clarify this in the text.

Line 68. Not clear what is meant by "scale-aware schemes". Does this refer to microphysics? Please provide citations.

Line 100-102. Not clear what is meant by "RRTMG considers the effects of aerosols on the effective sizes of hydrometeors". RRTMG accounts for radiative effects of both aerosols and hydrometeors, but not the effects of aerosols on hydrometeors. That is handled elsewhere, typically in the microphysics code.

Line 136. Before "less", insert "At pressures".

Line 141. Not clear what is meant by "cloud mass". Is it liquid water content?

Line 192. More description is needed here. The description of the model never discusses how turbulence or convection are represented.

Lines 227-230. Much more description is needed here. Surely more was changed than resolution. The 15 km and 35 km configurations must parameterize convection. How is that done?

Lines 247-248. "substantial decreases in theÂăcloud mass at the 15- and 35-km resolutions compared to the cloud mass in the simulations at the 500-m resolution". Since you refer to decreases, that suggests changes with aerosol, but I'm not sure if that is what you mean. You could mean there is less cloud mass at coarse resolution than at fine resolution. If you mean the latter, replace "are substantial decreases in the" with "is substantially less".

Line 267. As above, change "are decreases in" to "is less".

Line 271. At fine resolution?

Line 280. More discussion is needed here. At coarser resolution the updrafts are not resolved, so aerosol activation is poorly represented. If there is a cumulus parameterization, it probably lacks aerosol-aware microphysics.

Lines 283-285. If the GFS model lacks aerosol-aware physics then there would be little sensitivity to aerosol. The description of the GFS does not indicate any dependence on aerosol.

Line 291. Need discussion of how convection is represented in ARW at different resolutions. I assume the updraft mass flux in the course simulations is diagnosed from the convection scheme.

Lines 307-310. The difference could be due to poor parameterization of convection in both models. More information is needed.

Lines 311-312. Even if the GFS simulated the updraft mass flux correctly, it would likely still underestimate the sensitivity to the aerosol because it lacks the physics that drives the sensitivity.

Line 313. How do you get updraft speed from updraft mass flux? Cumulus parameterizations produce mass flux, but additional assumptions are needed to diagnose updraft speed. More detail is needed here.

Lines 323-339. Do the convection schemes in the models have any physics that would cause the updrafts to depend on aerosol?

Section 5.2. Excellent work and presentation!

Lines 468-473. Some discussion of the microphysics in the convection schemes used in the coarse resolution simulations would be helpful.

Lines 519-520. It should be noted here that in saturation adjustment schemes the condensed water does not depend on updraft velocity. And the coarse resolution models lack dependence of cumulus microphysics on supersaturation.

Lines 555-570. This is great discussion. Perhaps note that global models designed to represent cloud-aerosol interactions do use a subgrid updraft velocity for activation in

stratiform clouds, so they would exhibit less resolution dependence of clouds than the ARW model. See, e.g, Ghan et al. JGR 1997.

---

## Referee Comment (RC2) · Anonymous Referee #2 · 29 Jun 2017

This paper nicely demonstrates the role of spatial resolution and microphysics in determining differences between a model with high resolution and bin representation of microphysics compared to low resolution and bulk representation of microphysics. It should be published after clarification of the following and/or improvement in wording.

Many places use "resolutions" where I would have thought "resolution" was best English usages.

Line 71: Change "These" to This

Line 136: change "less than" to "above"

Lines 118 – 126: this cannot be the full description of ammonium sulfate sources and sinks, since it only describes the interaction of aerosol with clouds. What about nucleation from the gas phase production of sulfate? How is gas phase sulfate produced? Do you represent condensation onto existing aerosols? What about dry deposition loss?

Fig 1a,b: please increase size of rectangle, similar to 1c, d.

Model set up: What is used for boundary conditions for the CSRM? How do these boundary conditions compare to the incoming air in the GFS simulations?

Line 294: what are the deposition rates shown in Fig 9? This is not surface deposition, since the units are wrong.

Line 298-300: why do updraft mass fluxes increase with higher aerosol?

Line 526: what are "high-level" updrafts? At high altitude? Similar comment for low-level updrafts. You did not discuss this in the paper. (also only updraft mass flux is in figures).

---

## Author Comment (AC1) · 8 Sep 2017

**et al.**

**Anonymous Referee #1**

This manuscript is a bit of a mixed bag. I really like the analysis of the difference between the bin and bulk microphysics. The analysis of resolution dependence of the clouds and cloud-aerosol interactions is not so clear, as too little information is provided regarding the representation of deep convection at coarse resolution.

The illustrations are generally helpful, and the writing is mostly quite clear.

Lines 35-38. Is the comparison done at the same scale? We certainly wouldn't expect a coarse resolution simulation to produce the same updraft intensity as a fine resolution simulation if they aren't compared at the same scale. So I'm withholding judgement on this conclusion until I know more. Perhaps need to clarify this in the text.

As described in Section 4.2, an identical domain for each of the Seoul and Houston cases is applied to both the CSRM and GFS simulations. Stated differently, the spatial scale or the extent of the analysis area is identical between the CSRM and GFS simulations, although the number of grid points in the area or the domain is different between the CSRM and GFS simulations due to differences in resolutions between those simulations. To clarify the point here, the following is added:

(LL235-238 on p12)

Stated differently, the spatial scale or the extent of the analysis area is identical between the CSRM simulations and the GFS simulations, although the number of grid points in the area or the domain is different between the CSRM simulations and the GFS simulations due to differences in resolution between those simulations.

Line 68. Not clear what is meant by "scale-aware schemes". Does this refer to microphysics? Please provide citations.

Note that the traditional cumulus parameterizations are limited by the issue of scale separation (Yano, 2012). Due to this limitation, the traditional cumulus parameterizations can only be used in coarse

resolutions which are coarse than ~ 50 km and cannot be used in find resolutions which are finer than ~ 50 km. Nowadays, as mentioned in text, many NWP models start to adopt resolutions finer than ~ 50 km and due to this, to replace the traditional cumulus parameterizations, the scale-aware or scale-free schemes which can be used for any resolutions whether they are coarser than ~ 50 km or not have been developed. Since the scale-aware schemes are designed to replace the traditional cumulus parameterizations, the scale-aware schemes basically represent sub-grid-scale convective dynamic processes (e.g., cloud-scale updrafts and downdrafts) like the traditional cumulus parameterizations.

To clarify the point here, the following is added:

(LL62-65 on p3-4)

These scale-aware schemes, which represent sub-grid-scale dynamic processes (e.g., cloud-scale updrafts and downdrafts) that are associated with cloud convection as the traditional cumulus parameterizations do, are designed to be applied to the increased resolution in the NWP models.

We already provided citations about the scale-aware schemes in LL59-62 on p3 as follows:

Motivated by this, scale-aware cumulus parameterization schemes (e.g., Bogenschutz and Krueger, 2013; Thayer-Calder et al., 2015; Griffin and Larson, 2016) are being implemented into these models of different resolutions for better representation of clouds and precipitation.

Reference:

Yano, 2012, What is Scale Separation?: A Theoretical Reflection, obtainable at http://convection.zmaw.de/fileadmin/user_upload/convection/Convection/COST_Documents/Basic_Pa rameterization_Concepts_and_Issues/What_is_Scale_Separation___A_Theoretical_Reflection.pdf

Line 100-102. Not clear what is meant by "RRTMG considers the effects of aerosols on the effective sizes of hydrometeors". RRTMG accounts for radiative effects of both aerosols and hydrometeors, but not the effects of aerosols on hydrometeors. That is handled elsewhere, typically in the microphysics code.

We checked the code and found that the effective size of hydrometeors is calculated in the microphysics scheme adopted and then the calculated size is transferred to the RRTMG scheme for the calculation of the effects of clouds on radiation with the consideration of the effective size. To clarify this, the corresponding text is revised as follows:

(LL101-104 on p5-6)

The effective sizes of hydrometeors, which vary with varying aerosol properties, are calculated in a microphysics scheme that is adopted by this study and described below and the calculated sizes are transferred to the RRTMG. Then, the effects of the effective sizes of hydrometeors on radiation are calculated in the RRTMG.

Line 136. Before "less", insert "At pressures".

Done.

Line 141. Not clear what is meant by "cloud mass". Is it liquid water content?

Cloud mass in the scheme of Moorthi et al. (2001) is represented by cloud liquid content or cloud ice conent in g m$^{-3}$, depending on temperature. Here, cloud liquid represents droplets and cloud ice represents ice crystals. To clarify this, the following is added:

(LL156-158 on p8)

Here, cloud mass is represented by cloud liquid content (CLC) or cloud ice content (CIC), depending on temperature, and cloud liquid (cloud ice) represents droplets (ice crystals).

Line 192. More description is needed here. The description of the model never discusses how turbulence or convection are represented.

The CSRM explicitly resolves the cloud-scale convection and thus we do not use parameterizations (e.g., cumulus parameterizations) to represent the cloud-scale convection as described in text. To clarify this better, the relevant text is revised as follows:

(LL215-216 on p11)

Note that the cumulus parameterization scheme is not used in this domain where cloud-scale convection and associated convective rainfall generation are assumed to be explicitly resolved.

To indicate how the turbulence is represented, the following is added:

(LL104-106 on p6)

The ARW model considers the sub-grid-scale turbulence by adopting 1.5-order turbulence kinetic energy closure (Basu et al., 1998).

Lines 227-230. Much more description is needed here. Surely more was changed than resolution. The 15 km and 35 km configurations must parameterize convection. How is that done?

Here, for those 15-km and 35-km configurations or the repeated simulations with the 15-km and 35-km resolutions, we do not parameterize convection using schemes such as "cumulus parameterizations". We just want to identify the pure effects of resolutions on the simulations of clouds, precipitation, and their interactions with aerosol or we simply want to identify the pure errors caused by the use of the coarse resolutions via comparisons between the CSRM simulations with the 500-m or fine resolutions and the repeated simulations with the 15-km and 35-km resolutions or coarse resolutions. This is why we repeat the standard CSRM runs only by varying the resolutions. In case we apply the convection parameterization (which is not applied to the CSRM simulations with the 500-m resolution) to the repeated simulations with the 15-km and 35-km resolutions, comparisons between the CSRM simulations and repeated simulations are not able to isolate the effects of resolutions due to the fact

that not only resolutions but also the convection parameterization contributes to differences among those simulations.

By only varying the resolutions among the simulations and not applying the convection parameterizations to the repeated simulations with the coarse resolutions, we can say that differences between the CSRM simulations (with fine resolutions) and the repeated simulations (with coarse resolutions) are the errors in the simulations of clouds, precipitation, and their interactions with aerosol by taking the CSRM simulations as benchmark simulations, and the varying resolutions or the coarse resolutions are the only factor that produces the errors in the repeated simulations. To clarify the point here, the following is added:

(LL257-266 on p13-14)

To isolate the effects of resolution on the simulations of clouds, precipitation, and their interactions with aerosols, only resolution varies among the CSRM runs at the fine resolution and the repeated runs at the coarse resolutions here and these runs have an identical model setup except for resolution. For the identical setup, as an example, we do not apply the convection parameterizations (e.g., cumulus parameterizations) to the repeated runs, since the convection parameterizations are not applied to the CSRM runs. Hence, cloud variables (e.g., the updraft speed) are not diagnosed by convection parameterizations but predicted in both the CSRM runs and the repeated runs. With the identical setup except for resolution, the comparisons between the CSRM simulations and the repeated simulations can isolate the pure effects of the use of coarse resolution on clouds, precipitation, and their interactions with aerosol.

Note that the GFS simulation uses not only resolutions similar to those in the repeated simulations but also the convection parameterizations or cumulus parameterizations to represent the sub-grid convection.  Interestingly, the GFS simulation produces results which are similar to those in the repeated simulations with the 15-km and 35-km resolutions and are very different from those in the CSRM simulations with the 500-m resolutions despite the use of the convection parameterizations. This indicates that the use of the convection parameterizations, whose purpose is to correct the errors caused by the use of coarse resolutions and then to produce results similar to those in the simulations with the fine resolutions such as the CSRM simulations, does not correct the errors well.  This problem with the convection parameterizations is discussed in text.

Lines 247-248. "substantial decreases in theˇacloud mass at the 15- and 35-km resolutions compared to the cloud mass in the simulations at the 500-m resolution". Since you refer to decreases, that suggests changes with aerosol, but I'm not sure if that is what you mean. You could mean there is less cloud mass at coarse resolution than at fine resolution. If you mean the latter, replace "are substantial decreases in the" with "is substantially less".

Done.

Line 267. As above, change "are decreases in" to "is less".

Done.

Line 271. At fine resolution?

As described in Section 4.1 and in text in other sections such as that in LL 227-228 (in the old manuscript), the CSRM simulations are by definition those performed with the 500-m resolution or the fine resolution. To clarify this, the corresponding text is revised as follows:

(LL301-304 on p15-16)

In Figures 5 and 6, satellite-observed LWP and IWP for both cases follow reasonably well their CSRM-simulated counterparts for the polluted scenario. This shows that the CSRM simulations, which are performed with the 500-m resolution, perform well and can thus represent benchmark simulations.

Line 280. More discussion is needed here. At coarser resolution the updrafts are not resolved, so aerosol activation is poorly represented. If there is a cumulus parameterization, it probably lacks aerosol-aware microphysics.

As mentioned in our response to the comment for the Lines 227-230, in the repeated simulations with the coarse resolutions, we do not use cumulus parameterizations for the reasons detailed in the response to the comment for the Lines 227-330. As detailed in the response, by not using cumulus parameterizations, we can isolate the pure effects of the use of the coarse resolutions on clouds, precipitation, and their interactions with aerosol.  Here, these pure effects include the effects of updrafts not resolved by the coarse resolutions on clouds, precipitation, and their interactions with aerosol via microphysical processes such as activation. As detailed in our response to the comment for the Lines 227-330, these pure effects are none other than the differences in results between the CSRM simulations with the find resolution and the repeated simulations with the coarse resolutions; note that these differences represent the pure errors caused by the use of the coarse resolutions (as detailed in our response to the comment for the Lines 227-330) and for example, associated updrafts not resolved and poorly represented activation. These differences are compared to differences between the CSRM simulations and the GFS simulation to evaluate how the cumulus parameterization in GFS works to minimize the errors. Here, the differences between the CSRM simulations and the repeated simulations with the coarse resolutions should act as a maximum extent of the errors when the cumulus parameterizations to correct the errors are not used. As the differences between the CSRM simulations and the GFS simulation become closer to those between the CSRM simulations and the repeated simulations, the cumulus parameterization used in the GFS simulation is considered to be working worse. Stated differently, if there are no differences in results between the CSRM simulations and the GFS simulation, the cumulus parameterization in the GFS simulation is considered to work perfect. Here, as mentioned in text, the CSRM simulations act as benchmark simulations and this is proven by comparisons between the CSRM simulations and observations.

In summary, here, the pure errors include those caused by updrafts not resolved and poorly represented activation as exemplified by the reviewer here and we quantify these errors via the comparisons

between simulations as described above and based on quantification, we provide a guideline by which the parameterizations in GFS can be developed in an efficient way; see our discussion related to this in the last five paragraphs of "summary and discussion"

Lines 283-285. If the GFS model lacks aerosol-aware physics then there would be little sensitivity to aerosol. The description of the GFS does not indicate any dependence on aerosol.

It is true that there is no "aerosol-aware physics" in the current GFS. However, in the text pointed out here, it is meant that even though aerosol-aware physics is implemented into GFS, it is likely that GFS shows the weak sensitivity (to increasing aerosol concentration) as shown in the ARW simulations at the coarse resolution.

The cumulus parameterization in the current GFS is not able to correct errors in variables such as updrafts to result in the similarity between the GFS simulation and the ARW simulations at the coarse resolutions. The ARW simulations which are equipped with the aerosol-aware physics and at the coarse resolutions demonstrate that when those errors are not corrected or updrafts are underestimated (as compared to the CSRM simulations), even the presence of "aerosol-aware physics" does not prevent the weak sensitivity to increasing aerosol concentrations. Hence, the current GFS, which does not correct errors in updrafts well or underestimates updrafts as discussed in "summary and discussion", is likely to show the weak sensitivity even though "aerosol-aware physics" is implemented into GFS.

To clarify the point here, the following is added:

(LL363-374 on p19)

Taking the sensitivity of updraft mass fluxes to increasing aerosol concentrations in the CSRM simulations as the benchmark sensitivity, the GFS simulations likely also underestimate the sensitivity, considering the similarity in results between the ARW simulations at the 15- and 35-km resolutions and the GFS simulations. Since the current GFS model does not consider pathways through which increasing aerosol concentrations interact with updraft mass fluxes, this probable underestimation of the sensitivity is even more likely. Note that the ARW simulations which are at the 15- and 35-km resolutions and underestimate updrafts themselves, even with the consideration of those pathways, result in the much weaker sensitivity at the coarse resolutions as compared to that in the CSRM simulations. Hence, even though those pathways are implemented into the GFS model, the underestimated updrafts in the GFS simulations are likely to result in the weak sensitivity, unless the cumulus parameterization which represents updrafts in the GFS model is further developed to prevent the underestimation of updrafts.

Line 291. Need discussion of how convection is represented in ARW at different resolutions. I assume the updraft mass flux in the course simulations is diagnosed from the convection scheme.

The convection scheme such as the cumulus parameterization is not used in the ARW simulations at all of the different resolutions due to the reasons which are elaborated in our response to the comment for the Lines 227-230. The discussion of how convection is presented in the ARW simulations at different resolutions is given as follows:

(LL257-266 on p13-14)

To isolate the effects of resolution on the simulations of clouds, precipitation, and their interactions with aerosols, only resolution varies among the CSRM runs at the fine resolution and the repeated runs at the coarse resolutions here and these runs have an identical model setup except for resolution. For the identical setup, as an example, we do not apply the convection parameterizations (e.g., cumulus parameterizations) to the repeated runs, since the convection parameterizations are not applied to the CSRM runs. Hence, cloud variables (e.g., the updraft speed) are not diagnosed by convection parameterizations but predicted in both the CSRM runs and the repeated runs. With the identical setup except for resolution, the comparisons between the CSRM simulations and the repeated simulations can isolate the pure effects of the use of coarse resolution on clouds, precipitation, and their interactions with aerosol.

Lines 307-310. The difference could be due to poor parameterization of convection in both models. More information is needed.

See our responses to the comments on Line 280 and Lines 283-285.

Lines 311-312. Even if the GFS simulated the updraft mass flux correctly, it would likely still underestimate the sensitivity to the aerosol because it lacks the physics that drives the sensitivity.

See our responses to the comment on Lines 283-285. Based on them and the fact that the CSRM simulations act as benchmark simulations that predict updrafts correctly and show the benchmark sensitivity to the aerosol, it is believed that in case the GFS simulation predicts the updraft mass flux correctly and is equipped with "aerosol-aware physics" like the CSRM simulations, the GFS simulation is likely to produce a correct sensitivity.

Line 313. How do you get updraft speed from updraft mass flux? Cumulus parameterizations produce mass flux, but additional assumptions are needed to diagnose updraft speed. More detail is needed here.

In the ARW simulations at any resolutions, as explained in our responses above, cumulus parameterizations are not used. Instead, in those ARW simulations, the updraft speed itself is predicted by the ARW model. Then the updraft mass flux is obtained simply by multiplying the updraft speed with air density. To clarify the point here, the following is added:

(LL262-264 on p13-14)

Hence, cloud variables (e.g., the updraft speed) are not diagnosed by convection parameterizations but predicted in both the CSRM runs and the repeated runs.

(LL324-325 on p17)

Updraft mass fluxes are obtained by multiplying the predicted updraft speed by air density.

Lines 323-339. Do the convection schemes in the models have any physics that would cause the updrafts to depend on aerosol?

As elaborated in our responses above, the convection schemes are not used in the ARW simulations at any resolutions in this paper. Instead, based on results from those simulations, the discussion about the convection schemes with the dependence on aerosol is given in "summary and discussion"; see the last paragraph in the paper for the discussion.

Section 5.2. Excellent work and presentation!

Lines 468-473. Some discussion of the microphysics in the convection schemes used in the coarse resolution simulations would be helpful.

As elaborated in our responses above, the convection schemes are not used in the ARW simulations at any resolutions in this paper.

Lines 519-520. It should be noted here that in saturation adjustment schemes the condensed water does not depend on updraft velocity. And the coarse resolution models lack dependence of cumulus microphysics on supersaturation.

We believe that saturation adjustment calculates the amount of water vapor that should be condensed based on the predicted or diagnosed updrafts and the associated level of saturation. Even in saturation adjustment, stronger updrafts produce a lower level of saturation for a given amount of water vapor to result in more condensed water vapor that affects associated microphysical processes, as in schemes that predict supersaturation. However, in saturation adjustment, the entire amount of water vapor that is determined to be condensed is removed from the atmosphere within one time step, which is different from those schemes that predict supersaturation or the supersaturation prediction; Tao et al. (1989) is one of classic papers on saturation adjustment and see this paper for the details of saturation adjustment. Hence, in saturation adjustment, condensed water depends on updrafts or updraft speed. Stated differently, even in saturation adjustment, the underestimation of the updraft intensity leads to the underestimation of condensed water and thus cloud mass. Based on this, whether the NWP models adopt saturation adjustment (and its impacts on microphysics) or supersaturation prediction (and its impacts on microphysics), the use of coarse resolutions in the NWP models, which results in the underestimation of updrafts, induces the underestimation of condensed water and cloud mass. As detailed in our responses to some of the comments above, whether the NWP models adopt "aerosol-aware physics (that considers the effect of updrafts and supersaturation (or the saturation level) on cumulus microphysics and aerosol impacts on the effect)" as phrased by the reviewer above, the NWP models are likely to underestimate cloud variables (e.g., updrafts, condensed water and cloud mass) and the sensitivity of cloud variables to increasing aerosol concentrations based on the ARW simulations.

Text pointed out here is revised as follows to remove impression that the statement in this text is only applicable to the supersaturation prediction but not the saturation adjustment:

(LL577-579 on p29)

This study shows that the use of coarse resolution can cause an underestimation of the updraft intensity and thus condensation and deposition, which leads to an underestimation of the cloud mass.

Reference:

Tao, W. K., J. Simpson, and M. McCumber, 1989, An ice-water saturation adjustment, J. Appl. Meteor., 117, 231-235.

Lines 555-570. This is great discussion. Perhaps note that global models designed to represent cloud-aerosol interactions do use a subgrid updraft velocity for activation in stratiform clouds, so they would exhibit less resolution dependence of clouds than the ARW model. See, e.g, Ghan et al. JGR 1997.

As detailed in our discussion (LL540-554 in the old manuscript), although the GFS model is coupled to the sub-grid parameterizations or the convection scheme such as cumulus parameterizations that diagnose the subgrid variables such as the subgrid updraft speed, the GFS model produces the results similar to those in the ARW simulations at the coarse resolution. As detailed in our discussion (LL540-554 in the old manuscript), this means that the diagnosis of the subgrid variables (by the convection scheme) and the calculation of their impacts on microphysical processes such as activation do not work well in the GFS model. Associated with this, as detailed in our responses to some of the reviewer's comments above, although "aerosol-aware physics" as phrased by the reviewer here is implemented into GFS, GFS is likely to show the weak sensitivity (to increasing aerosol concentration) as shown in the ARW simulations at the coarse resolution. Stated differently, the GFS model (representing global models) which considers cloud-aerosol interactions is likely to produce the weak sensitivity or the underestimation of the sensitivity, unless the convection scheme is improved to remove the errors that are caused by the use of the coarse resolution and the subsequent incorrect diagnosis of the subgrid variables. This discussion about the GFS model (representing global models) is all about the GFS model at the coarse resolution. In this study, we assume that in case GFS model adopts the resolution as fine as in the CSRM simulations, the GFS model does not have to use the convective schemes as in the CSRM simulations, since cloud variables such as updrafts are considered to be explicitly resolved at the fine resolution. Considering that the dependence of model results on microphysics parameterizations is very small as compared to that on resolutions as discussed in Section 5.3, we assume that the GFS model (representing the global models) produce results (including the sensitivity) similar to those in the CSRM as long as the GFS model uses the resolution as fine as in the CSRM and thus explicitly resolves updrafts as in the CSRM, although there can be differences in physics parameterizations between the GFS and the CSRM. Based on this assumption and the fact that at the coarse resolution, results are similar between the GFS model and the ARW model, we believe that the variation of results (including the sensitivity) among the ARW simulations with a transition from a fine resolution to a coarse resolution is likely to be similar to that variation among the GFS simulations with that transition.

---

## Author Comment (AC2) · 8 Sep 2017

This paper nicely demonstrates the role of spatial resolution and microphysics in determining differences between a model with high resolution and bin representation of microphysics compared to low resolution and bulk representation of microphysics. It should be published after clarification of the following and/or improvement in wording.

Many places use "resolutions" where I would have thought "resolution" was best English usages.

We went through text and replaced "resolutions" with "resolution" when needed.

Line 71: Change "These" to This

Done.

Line 136: change "less than" to "above"

Following the other reviewer's comment, "less than" is replaced with "At pressures less than"

Lines 118 – 126: this cannot be the full description of ammonium sulfate sources and sinks, since it only describes the interaction of aerosol with clouds. What about nucleation from the gas phase production of sulfate? How is gas phase sulfate produced? Do you represent condensation onto existing aerosols? What about dry deposition loss?

In this study, we focus on interactions among aerosol, clouds, and precipitation but not on aerosol physics and chemistry. Stated differently, this study aims to examine errors and mechanisms that govern those errors in the simulations of aerosol-cloud-precipitation interactions themselves by the NWP models as stated in "introduction". Thus, the examination of errors and associated mechanisms in the simulations of aerosol physics and chemistry by the NWP models is out of scope of this study. Based on this, we do not explicitly simulate aerosol physics and chemistry and we prescribe aerosol physical and chemical properties. To clarify the points here, the following is added:

(LL123-130 on p7)

As stated in introduction, this study focuses on the uncertainties or errors in the simulations of clouds, precipitation, and CAPI themselves. This means that the examination of the uncertainties in the simulations of aerosol physics and chemistry is out of scope of this study. Hence, in this study, instead of simulating aerosol physics and chemistry explicitly, initial aerosol physical and chemical properties (i.e., aerosol chemical composition and size distribution) are prescribed. Then, aerosol size distribution (or aerosol number concentration in each size bin) evolves only through cloud processes but not through aerosol physical and chemical processes. During the evolution, the prescribed aerosol composition is assumed not to vary.

Fig 1a,b: please increase size of rectangle, similar to 1c, d.

Done.

Model set up: What is used for boundary conditions for the CSRM? How do these boundary conditions compare to the incoming air in the GFS simulations?

For the ARW simulations (including the CSRM simulations), we use open lateral boundary conditions and hence, the synoptic conditions are advected into and out of the domain by air through the boundary of the domain. This emulates the situation in the GFS simulations where the synoptic conditions are advected into and out of an area (corresponding to the domain in the ARW simulations) by air through the border between the area and places outside the area. In the ARW simulations, the synoptic conditions are derived from the National Centers for Environmental Prediction GFS final (FNL) analysis. The FNL analysis is based on environmental conditions that are produced by GFS and thus there are basically no differences in the synoptic conditions between those advected into and out of the domain in the ARW simulations by air and those advected into and out of the area (corresponding to the domain in the ARW simulations) in the GFS simulations by air.

In summary, the domain in the ARW simulations and the area (corresponding to the domain in the ARW simulations) in the GFS simulations experience an identical synoptic condition. The advection of the synoptic condition into and out of the domain by air in the ARW simulations is through the boundary of the domain, which is enabled by the use of the open boundary conditions and emulates the advection of the synoptic condition into and out of the area in the GFS simulations through the border between the area and places outside the area.

To clarify the point here, the following is added:

(LL198-205 on p10-11)

Initial and boundary conditions, which represent the synoptic features, for the control run are derived from the National Centers for Environmental Prediction GFS final (FNL) analysis. Since the FNL analysis is based on environmental conditions that are produced by the GFS model and thus for each of the cases, there are basically no differences in the synoptic condition between the CSRM simulations and the GFS simulations that are described in the following Section 4.2. The open lateral boundary condition is adopted in the control run. This enables the advection of the synoptic condition into and out of a domain in the CSRM simulations to occur through the boundary of the domain, which emulates the advection in the GFS simulations.

Line 294: what are the deposition rates shown in Fig 9? This is not surface deposition, since the units are wrong.

In this paper, condensation, evaporation, and deposition occur on the surface of hydrometeors. Condensation and evaporation occur on the surface of drops, while deposition occurs on the surface of solid hydrometeors such as ice crystals; deposition is, by definition in microphysics, the diffusion of water vapor onto solid hydrometeors such as ice crystals in clouds. In this paper, condensation rate and evaporation rate are defined as the rates of changes in drop mass (or liquid mass) in a unit volume of air and for a unit time due to condensation and evaporation, respectively, following the conventional definition of condensation and evaporation rates in cloud community. Deposition rate is defined as the rate of changes in the mass of solid hydrometeors (or ice mass) in a unit volume of air and for a unit time due to deposition, following the conventional definition of deposition rate in cloud community. To clarify this, the following is added:

(LL331-333 on p17)

Here, condensation and deposition rates are defined as the rates of changes in liquid mass and ice mass in a unit volume of air and for a unit time due to condensation and deposition on the surface of hydrometeors, respectively.

(LL414-416 on p21)

Here, evaporation rate is defined as the rate of changes in liquid mass in a unit volume of air and for a unit time due to evaporation on the surface of hydrometeors.

Line 298-300: why do updraft mass fluxes increase with higher aerosol?

As stated in text (LL331-335) in the old manuscript, aerosol-induced invigoration of convection through aerosol-induced increases in freezing or aerosol-induced intensification of gust fronts is the main mechanism behind aerosol-induced increases in updraft mass fluxes or the intensity of updrafts. To provide more detailed information on aerosol-induced invigoration of convection, the following is added:

(LL336-349 on p17-18)

Increasing aerosol concentrations alter cloud microphysical properties such as drop size and autoconversion. Aerosol-induced changes in autoconversion in turn increase cloud-liquid mass as a source of evaporation and freezing. Numerous studies (e.g., Khain et al., 2005; Seifert and Beheng, 2006; Tao et al., 2007, 2012; van den Heever and Cotton, 2007; Storer et al., 2010; Lee et al., 2013, 2017) have shown that aerosol-induced increases in cloud-liquid mass and associated increases in freezing of cloud liquid enhance the freezing-related latent heating and thus parcel buoyancy, and this invigorates convection or increases updraft mass fluxes. Those studies have also shown that the aerosol-induced increases in cloud-liquid mass and associated increases in the evaporation of cloud liquid enhance the evaporation-related latent cooling and thus negative buoyancy. This intensifies downdrafts and after reaching the surface, the intensified downdrafts spread out toward the surrounding warm air to form intensified gust fronts and then, to uplift the warm air more strongly. More strongly uplifted warm air leads to invigorated convection or increased updraft mass fluxes. These freezing- and evaporation-related invigoration mechanisms are operative to induce the aerosol-induced enhancement of updraft mass fluxes, condensation, and deposition in this study.

Line 526: what are "high-level" updrafts? At high altitude? Similar comment for lowlevel updrafts. You did not discuss this in the paper. (also only updraft mass flux is in figures).

Discussion in the paragraph related to this comment is based on Figure 7 that shows the vertical distributions of the time- and domain-averaged updraft mass fluxes in the ARW simulations. Just want to note that updraft mass fluxes are obtained by multiplying the predicted updraft speed (or updrafts) with air density. Since there are negligible differences in air density among simulations at different resolutions, differences in updraft mass fluxes among those simulations are mostly caused by differences in the updraft speed or updrafts. This means that the qualitative nature of discussion about the differences in updraft mass fluxes among simulations is not different from that of discussion about the differences in the updraft speed or updrafts. Hence, discussion in paragraphs using the word "updrafts" can be replaced with discussion using the word "updraft mass fluxes". In the paragraph, high-level and low-level simply mean high-value and low-value, respectively.

To clarify our points here, the following is added:

(LL324-329 on p17)

Updraft mass fluxes are obtained by multiplying the predicted updraft speed by air density. Since there are negligible differences in air density among the ARW simulations, most of differences in updraft mass fluxes among the simulations are caused by differences in the updraft speed or updrafts. Those differences in air density are in general ~ two orders of magnitude smaller than those in the updraft speed or updrafts.

Based on our points here, the corresponding text is revised as follows:

(LL585-588 on p30)

When they are resolved with the use of high-resolution models, there are high-value averaged updrafts and associated variables, and their strong sensitivity but when they are not resolved in low-resolution models, there are low-value averaged updrafts and associated variables, and their weak sensitivity.